# Think in Latent, Explain in Language: Self-Explainable Latent Reasoning

**Dayuan Zhao** [1]  **Shengcao Cao** [1]  **Yu-Xiong Wang** [1]  **Liang-Yan Gui** [1]

## Abstract

Latent reasoning has emerged as a powerful alternative to text-based Chain-of-Thought (CoT), offering significant gains in computational efficiency by compressing verbose reasoning into compact embeddings. However, compressing reasoning into the latent space renders the thinking opaque, hindering its interpretability. Current methods present a stark trade-off: they either function as unexplainable "black boxes" (*e.g.*, Coconut), where the latent reasoning is not human-readable, or rely on separate post-hoc decoders for explainability (*e.g.*, Heima), introducing architectural overhead and decoupling the explanation from the actual reasoning process. In this work, we present a unified framework for Self-Explainable Latent Reasoning (SELR) that trains a single model to perform efficient and inherently explainable latent reasoning. Our core contribution is a novel multi-task training objective that optimizes for two goals simultaneously: (1) an Answer Loss that optimizes the latent reasoning trajectory to produce accurate final answers, and (2) a CoT Loss that explicitly trains the same model to decode its own latent representations back into human-understandable reasoning steps. This design ensures that generated latent representations are both task-effective and semantically interpretable, eliminating the need for external decoders. We validate the effectiveness of SELR on both Large Language Models (LLMs) and Vision-Language Models (VLMs), demonstrating that SELR achieves superior token efficiency and accuracy compared to baselines, while uniquely providing self-contained explainability without auxiliary models. Project page is available at https://jasondayuan.github.io/SELR/.

[1]Siebel School of Computing and Data Science, University of Illinois Urbana-Champaign, Urbana, Illinois, USA. Correspondence to: Dayuan Zhao <dayuan@illinois.edu>.

*Proceedings of the 43ʳᵈ International Conference on Machine Learning*, Seoul, South Korea. PMLR 306, 2026. Copyright 2026 by the author(s).

## 1. Introduction

Large Language Models (LLMs) have traditionally been trained to reason in the *language space*, where they elaborate their thinking process with textual Chain-of-Thought (CoT) (Wei et al., 2022). However, this discrete language space may not be the optimal medium for reasoning: Text-based CoT can be excessively verbose, with many tokens serving textual coherence rather than actual reasoning, and is fundamentally constrained by a fixed, discrete vocabulary. To overcome these limitations, a new paradigm of reasoning in a *continuous latent space* (Hao et al., 2025) has emerged: Models can reason more effectively by feeding the last hidden state (a "continuous thought") directly back into the model as the next input embedding, bypassing the discrete text generation step. This allows models to produce their thinking procedure with continuous tokens, which can encode more dense information as they are no longer constrained by the discrete embedding space. This paradigm enables unique synergy in Vision-Language Models (VLMs), as it bridges the modality gap between continuous visual embedding and discrete textual tokens, allowing the model to process visual information in its native continuous format. Consequently, latent space reasoning models have the potential to be significantly more *token-efficient* than their text-based counterparts.

Despite this promise, current latent space reasoning models face two significant challenges. The first is the **lack of supervision**. While humans can write textual CoT steps to serve as a supervision source for text-based reasoning models, we cannot supervise latent space models with the same approach, as there is no human-interpretable "ground truth" for such continuous thoughts. Prior efforts have explored methods like multi-stage curriculum learning (Hao et al., 2025) to gradually shift the thinking procedure from the language space to the latent space, but there is no generally agreed-upon approach for learning latent space reasoning effectively.

The second problem is the **lack of interpretability**. When a latent space reasoning model produces its thinking procedure in the latent space, there is no direct approach for us to know *what the model is "thinking" about*. Although we can probe the continuous thought tokens to find their most similar text tokens in the model's language space (Hao et al.,

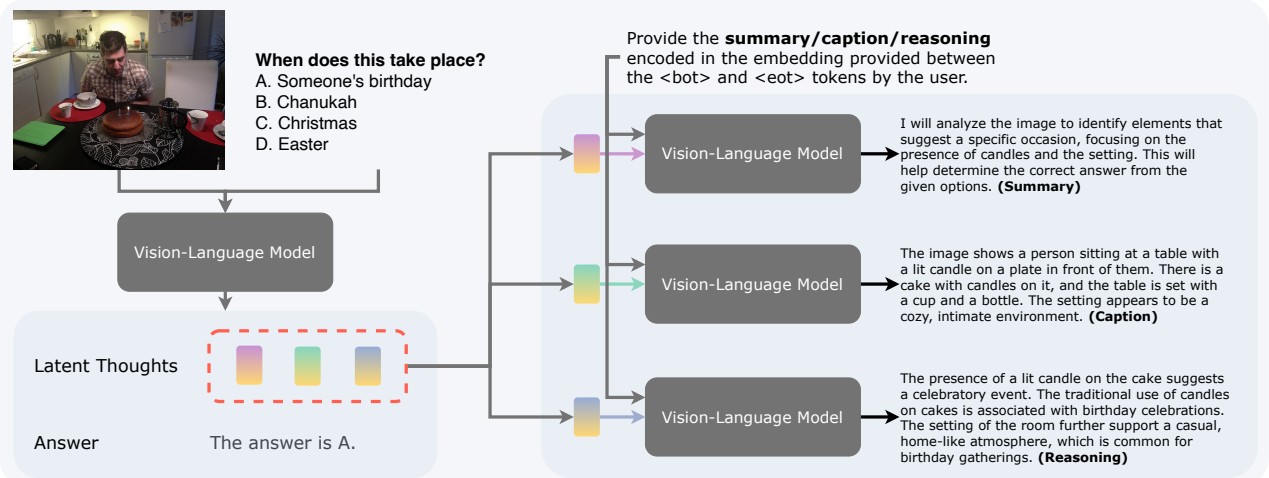

*Figure 1.* Demonstration of the SELR framework. **Left (Reasoning):** The image and question are provided as input to the VLM, which performs latent space reasoning to generate latent thoughts and a final answer. **Right (Decoding):** The generated latent thoughts are then provided as input back to the very same VLM, along with a decoding prompt. The VLM then decodes its own latent thoughts, presenting the information encoded within them as human-readable text.

2025), this only explains one token at a time and is poorly human-readable. Another explored approach (Shen et al., 2025a) achieves explainability by training a separate, post-hoc decoder model, but this disjoint architecture introduces significant parameter overhead and risks decoupling the explanation from the actual reasoning logic, as the explainer is distinct from the reasoner. While token-efficient thinking is desired, we lose the crucial ability to conveniently understand the model's reasoning path when necessary.

To address these challenges, we propose a unified framework, Self-Explainable Latent Reasoning (SELR), that targets both supervision and interpretability simultaneously. Our central design is a **multi-task learning objective** that trains a single model to be both an efficient latent space *reasoner* and its own latent-to-language *translator*. Specifically, the model is trained to optimize two goals concurrently: (1) an **Answer Loss** that guides the latent reasoning to produce the correct final answer, ensuring high reasoning performance, and (2) a **CoT Loss** that explicitly trains the *same model* to decode its own latent representations back into human-understandable reasoning steps. A demonstration of this process is provided in Figure 1.

This multi-task approach provides a synergistic solution to both aforementioned challenges. The CoT Loss provides a rich, text-based supervisory signal that guides the formation of the latent thoughts, addressing the supervision challenge. Simultaneously, it forces the model to learn latent representations that are inherently aligned with human logic, thus solving the interpretability problem. The model learns to generate *token-efficient* thoughts that are not only *effective*

for solving the task but also inherently *explainable*.

We conduct comprehensive experiments to explore the best practices for training such models. We first run extensive ablations and comparative experiments on lightweight LLMs using small-scale, text-only datasets to identify the most effective strategies. We then demonstrate that these strategies generalize successfully to the complex VLM domain. In particular, we show that a state-of-the-art VLM (Qwen2.5-VL (Bai et al., 2025)) fine-tuned with SELR achieves simultaneous gains in both efficiency and accuracy by generating significantly fewer tokens and attaining higher scores across multiple VLM benchmarks, while demonstrating strong interpretability by decoding its own continuous thoughts.

Our key contributions are threefold:

1. We propose a novel framework SELR that, *for the first time*, trains a single model to reason efficiently in latent space while also being able to translate its latent thoughts into human-readable text, solving the concurrent problems of supervision and interpretability.

2. We are *the first* to successfully apply this explainable latent reasoning paradigm to VLMs, demonstrating that a state-of-the-art model can achieve simultaneous improvements in both reasoning accuracy and token efficiency.

3. We provide a comprehensive analysis of training strategies, establishing a set of generalizable best practices for developing efficient and explainable latent space reasoning models.

## 2. Related Work

**Chain-of-Thought Reasoning.** The Chain-of-Thought (CoT) reasoning paradigm begins with prompting techniques that elicit step-by-step reasoning in LLMs, either through few-shot examples (Wei et al., 2022) or simple prompts (Kojima et al., 2022). This approach is enhanced by decoding strategies such as self-consistency (Wang et al., 2023) and generalized by search-based structures like Tree of Thoughts (Yao et al., 2023). The CoT framework is also adapted for vision-language tasks (Zhang et al., 2024). CoT has shifted from prompting to internalization, using process supervision to train models like OpenAI o1 (OpenAI, 2024b) or based on reinforcement learning like DeepSeek R1 (DeepSeek-AI, 2025).

**Latent Space Reasoning.** To overcome the computational inefficiency of explicit CoT, the seminal Coconut framework (Hao et al., 2025) moves reasoning to a continuous latent space by training a model to "internalize" text-based CoT traces, feeding the last hidden state back as a "continuous thought." Latent space reasoning allows the model to perform advanced, non-deterministic reasoning like a latent Breadth-First Search (Zhu et al., 2025a) and has been extended in various directions, such as iterative refinement (Geiping et al., 2025), energy minimization (Gladstone et al., 2025), or pre-training integrating (Zhu et al., 2025b). Follow-up methods of Coconut, such as CODI (Shen et al., 2025b), CoLaR (Tan et al., 2025), and HRPO (Yue et al., 2025) improve efficiency by compressing CoT using strategies like curriculum learning or reinforcement learning. However, the "black box" interpretability problem persists. While the Heima framework (Shen et al., 2025a) addresses this, it requires training separate models: an encoder to compress the thoughts and a decoder to interpret them. Our SELR method solves this by training a single, unified model to be both an efficient reasoner and its own interpreter.

## 3. Approach

In this section, we begin by establishing the preliminaries of the VLM generation process. Building on this foundation, we introduce the core optimization objectives of SELR, specifically the Answer Loss and CoT Loss. Finally, we detail the complete training framework, presenting both our single-stage and multi-stage curriculum strategies.

### 3.1. Preliminary: VLM Generation

Given an image $I$ and a text token sequence $X_t$, the generation process of a VLM can be described as follows:

$$E_v = \mathcal{P}(\mathcal{V}(I)), E_t = \mathcal{E}(X_t),$$
$$H = \text{LLM}([E_v; E_t]),$$
$$p(x_{l+1}|I, X_t) = \text{Softmax}(W h_l),$$

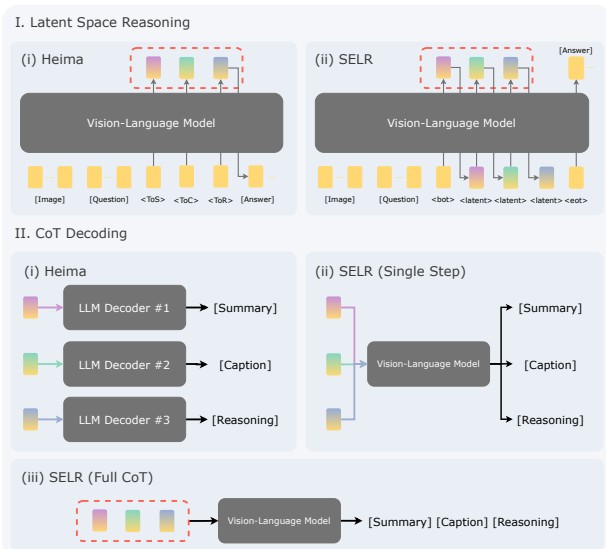

*Figure 2.* Visualization of our SELR framework, which consists of two stages. **(I) Latent Space Reasoning:** The model first generates a sequence of latent thoughts and the final answer, given the image and question. Heima utilizes special "Thinking of Summary/Caption/Reasoning" tokens for latent reasoning, whereas SELR generates latent thoughts sequentially, similar to Coconut. **(II) CoT Decoding:** The generated latent thoughts are then decoded. Heima requires training multiple, separate LLM decoders for each reasoning step, while **SELR** trains the *single, original model* to be its own translator, using the very VLM that generates the latent space reasoning to decode its own latent thoughts. For SELR methods trained with Single Step Loss, we decode the reasoning steps one-by-one from each latent thought; for SELR methods trained with Full CoT Loss, we decode the entire CoT with all latent thoughts at once.

where $\mathcal{V}$ is the vision encoder (*e.g.*, a ViT (Dosovitskiy et al., 2021)); $\mathcal{P}$ is the projection layer; $\mathcal{E}$ is the embedding matrix; $E_v \in \mathbb{R}^{l_v \times d}$ is the sequence of image embeddings; $E_t \in \mathbb{R}^{l_t \times d}$ is the sequence of text token embeddings; $H \in \mathbb{R}^{l \times d}$ is the sequence of final hidden states, where $l = l_v + l_t$; $h_l$ is the final hidden state corresponding to position $l$, which is the last position of the input sequence; $W$ is the parameter of the language model head.

### 3.2. Learning Objective

The core design of SELR is that a model's internal reasoning state should be both capable of solving the task and *self-explainable*. To achieve this, we employ a multi-task learning objective on a dataset of image-question pairs annotated with ground-truth CoT and final answers. Each instance can be defined as:

$$(I, X_q, \{X_{\text{CoT}_k}\}_{k=1}^K, X_a),$$

where $I$ is the image, $X_q$ is the question, $X_{\text{CoT}_k}$ is the $k$-th reasoning step, and $X_a$ is the answer.

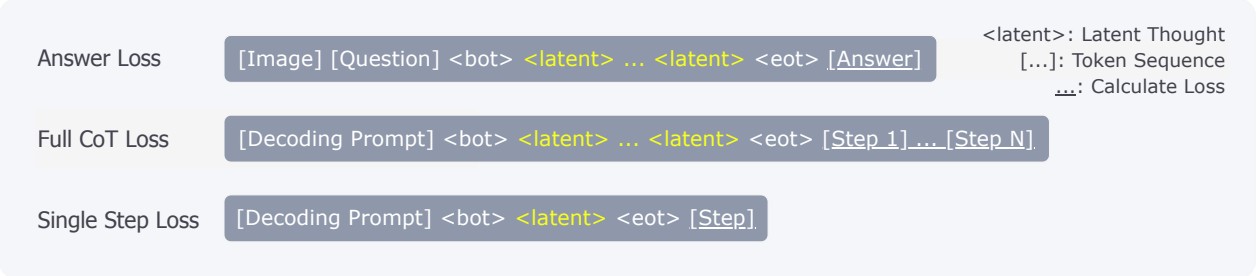

*Figure 3.* Illustration of the SELR multi-task loss objectives. The **Answer Loss** is computed on the answer tokens, conditioned on the image, question, and latent thoughts. The **Full CoT Loss** and **Single Step Loss** trains the model to decode its latent thoughts back into human-readable text, calculated on the entire CoT sequence and the sampled CoT step respectively.

Our optimization objective is the weighted sum of two loss components:

1. **Answer Loss:** Cross-entropy loss of answer tokens $X_a$ given image $I$, question $X_q$, and latent reasoning.

2. **CoT Loss:** Cross-entropy loss of the entire CoT sequence $X_{\text{CoT}} = [X_{\text{CoT}_1}, \ldots, X_{\text{CoT}_K}]$ or a single CoT step $X_{\text{CoT}_i}$ given the latent reasoning.

As an additional note, although this formulation is presented for VLMs, it is adaptable to LLM training by simply removing the image-related components (*e.g.*, $I$ and $E_v$).

### 3.2.1. LATENT SPACE REASONING GENERATION

To define the Answer Loss and the CoT Loss, we first specify the generation process for latent space reasoning, as both losses are conditioned upon it. Unlike standard generation, which projects final hidden states into a distribution over the vocabulary to sample discrete tokens, latent space reasoning feeds the final hidden state directly back as the next input embedding, effectively bypassing the language model head, as shown in Figure 2.

SELR uses two special tokens, `<bot>` and `<eot>`, to mark the beginning and end of the latent reasoning process. The full generation process is detailed in Algorithm 1.

The process begins by initializing an embedding sequence $E$ with the image features $\mathcal{P}(\mathcal{V}(I))$, the question embeddings $\mathcal{E}(X_q)$, and the `<bot>` embedding in order. The model then performs $l_{\text{latent}}$ autoregressive steps: in each step, the model takes the current sequence $E$ as input, and the final hidden state from the last position $H[-1, :]$ is appended back to $E$ to serve as the input for the next step. The final output is a sequence of latent thoughts $E_{\text{latent}} \in \mathbb{R}^{l_{\text{latent}} \times d}$, which used to compute losses afterwards.

**Thinking Budget.** A key design choice in our framework is the length of the latent reasoning sequence $l_{\text{latent}}$. We

---

**Algorithm 1** Latent Space Reasoning Generation

**Require:** Image $I$, Question $X_q$, Latent length $l_{\text{latent}}$
1: **Initialize:** $E \leftarrow [\mathcal{P}(\mathcal{V}(I)); \mathcal{E}(X_q); \mathcal{E}(\texttt{<bot>})]$
2: **Initialize:** $E_{\text{latent}} \leftarrow []$
3: **for** $k = 1$ **to** $l_{\text{latent}}$ **do**
4: $\quad H \leftarrow \text{LLM}(E)$ $\qquad\qquad\qquad$ ▷ Forward pass
5: $\quad h_{\text{last}} \leftarrow H[-1, :]$ ▷ Final hidden state of last position
6: $\quad E \leftarrow [E; h_{\text{last}}]$ $\qquad$ ▷ Append state as next input
7: $\quad E_{\text{latent}} \leftarrow [E_{\text{latent}}; h_{\text{last}}]$ $\qquad$ ▷ Store latent thought
8: **end for**
9: **Return** $E_{\text{latent}}$

---

explore two strategies:

1. **Fixed Length:** $l_{\text{latent}}$ is a constant hyperparameter which remains the same for all data instances.

2. **Variable Length:** $l_{\text{latent}}$ is set to match the number of ground-truth reasoning steps $K$ in the data instance. Therefore, its value varies from one instance to another.

For models trained with a fixed latent length, we simply terminate the latent generation loop after $l_{\text{latent}}$ steps at inference time. For models trained with a variable latent length, a fixed budget is still required at inference as the ground-truth $K$ is unavailable. We have explored training the model to autonomously terminate its latent reasoning, but it was found that this objective is significantly hard to train. Therefore, we apply a predefined fixed budget for models trained with variable latent length during inference as well.

### 3.2.2. ANSWER LOSS

The Answer Loss is a standard cross-entropy loss over the ground-truth answer tokens $X_a$, given the image $I$, question $X_q$, and latent thoughts $E_{\text{latent}}$ as context, as depicted in Figure 3. This loss supervises the model to be capable of arriving at the correct answer with latent space reasoning.

To compute this loss, we first construct the embedding sequence $E_{\text{ans}}^{\text{prompt}}$, which contains the original inputs, the full latent thought sequence, and the <eot> token:

$$E_{\text{ans}}^{\text{prompt}} = [\mathcal{P}(\mathcal{V}(I)); \mathcal{E}(X_q); \mathcal{E}(\texttt{<bot>}); E_{\text{latent}}; \mathcal{E}(\texttt{<eot>})]$$

The Answer Loss $L_{\text{ans}}$ is defined as the average negative log-likelihood over the answer tokens:

$$L_{\text{ans}} = -\frac{1}{|X_a|} \log p(X_a | E_{\text{ans}}^{\text{prompt}})$$

where $\log p(X_a | E_{\text{ans}}^{\text{prompt}})$ is the sum of the log-likelihoods of the tokens in $X_a$, conditioned on both $E_{\text{ans}}^{\text{prompt}}$ and other preceding tokens in the answer sequence.

### 3.2.3. CoT Loss

The CoT Loss is designed to provide supervision and ensure the interpretability of the latent thoughts. As introduced earlier, CoT Loss is the loss of the entire CoT sequence $X_{\text{CoT}}$ or a single CoT step $X_{\text{CoT}_i}$ given the model's latent space reasoning. The former type is named **Full CoT Loss**, and the latter is named **Single Step Loss**.

1. **Full CoT Loss (Global Alignment).** Provided with a decoding prompt $X_{\text{dec}}$ (*e.g.*, "Provide the reasoning process encoded in the embeddings provided between the <bot> and <eot> tokens by the user.") and the entire latent thought sequence $E_{\text{latent}}$, the Full CoT Loss is the loss of the entire textual CoT $X_{\text{CoT}}$, as shown in Figure 3.

$$E_{\text{dec,full}}^{\text{prompt}} = [\mathcal{E}(X_{\text{dec}}); \mathcal{E}(\texttt{<bot>}); E_{\text{latent}}; \mathcal{E}(\texttt{<eot>})]$$

$$L_{\text{dec,full}} = -\frac{1}{|X_{\text{CoT}}|} \log p(X_{\text{CoT}} | E_{\text{dec,full}}^{\text{prompt}})$$

2. **Single Step Loss (Fine-Grained Alignment).** Provided with the decoding prompt $X_{\text{dec}}$ and a single latent thought from the entire latent thought sequence $E_{\text{latent}_i}$, the Single Step Loss is the loss of the corresponding textual reasoning step $X_{\text{CoT}_i}$, as shown in Figure 3. Note that this loss can only be used with variable latent reasoning length, due to the one-to-one correspondence between textual reasoning steps and latent thoughts.

$$E_{\text{ans,step}}^{\text{prompt}} = [\mathcal{E}(X_{\text{dec}}); \mathcal{E}(X_{bot}); E_{\text{latent}_i}; \mathcal{E}(X_{eot})]$$

$$L_{\text{dec,step}} = -\frac{1}{|X_{\text{CoT}_i}|} \log p(X_{\text{CoT}_i} | E_{\text{ans,step}}^{\text{prompt}})$$

It is worth noting that the reasoning step used in each training step is sampled from a distribution, *i.e.*, for a data instance with $K$ reasoning steps, we sample the step index $i$ from a probability distribution over $\{1, 2, \ldots, K\}$. We consider two types of distributions: (1) **Uniform**, where each step has equal probability $p(i) = 1/K$; and (2) **Exponential**, where the probability doubles at each step, defined as $p(i) = 2^i/(2^{K+1} - 1)$ for $i \in \{1, \ldots, K\}$.

Crucially, during CoT decoding, the model is *not* shown the original image or question—only the latent thoughts. This self-regularization forces the latent thoughts to become a self-contained information bottleneck, carrying all necessary context for the solution.

### 3.3. Learning Stages

According to the number of stages involved during training, we categorize our training frameworks into two types: (1) **Single-Stage Methods**, which optimize a weighted sum of the aforementioned losses throughout training, and (2) **Multi-Stage Methods**, which employ a curriculum of sequential stages with distinct loss configurations.

#### 3.3.1. SINGLE-STAGE METHODS

Based on previous discussion, we can see that there are two design options for single-stage methods: (1) the latent reasoning length $l_{\text{latent}}$, *i.e.*, using fixed or variable latent length, and (2) the specific combination of losses to optimize.

For both LLM and VLM, we define the single-stage SELR method as using fixed latent length and the weighted sum of Answer Loss and Full CoT Loss as the objective. Both losses have a weight of 1.0, thus the objective is effectively the sum of both losses.

#### 3.3.2. MULTI-STAGE METHODS

We observed that the performance of the single-stage method was limited, particularly for LLMs. To improve the model's ability to learn effective latent representations, we introduce a multi-stage training strategy, inspired by the curriculum learning approach of Coconut (Hao et al., 2025). This method consists of multiple stages: an initial alignment stage, followed by a refinement stage. The specific implementation differs between LLMs and VLMs.

For LLM, the first stage trains the model with variable latent length and an objective that sums the Answer Loss and the Single Step Loss. This aligns the latent thoughts with the ground-truth reasoning steps. The second stage then trains the model with fixed latent length, and switches the objective to the sum of Answer Loss and Full CoT Loss.

For VLM, the first stage is identical to the LLM method. In the second stage, the model is also trained with fixed latent length. However, unlike the LLM method, we continue to use the sum of the Answer Loss and the Single Step Loss instead of switching to the Full CoT Loss. This design is intentional. The ground-truth reasoning steps in the VLM dataset, LLaVA-CoT-100k (Xu et al., 2025), are

long and information-dense, often spanning multiple sentences, whereas the LLM dataset's steps are single math equations. In support of this design choice, we find that the decoding quality degrades if we use Full CoT Loss for the second stage in Section 4.2.3. Crucially, every instance in LLaVA-CoT-100k provides exactly three fixed stages (summary, caption, reasoning), so the VLM latent length is effectively a constant, i.e., $l_{\text{latent}} = K = 3$. This preserves the one-to-one correspondence between latent thoughts and reasoning steps required by Single Step Loss even in the second stage, and the fixed ordering (summary $\rightarrow$ caption $\rightarrow$ reasoning) allows each latent position to specialize in encoding a specific type of information. See Appendix K for a detailed discussion on step ordering.

# 4. Experiments

In this section, we first introduce the experiment setup, including the datasets, implementation details, baselines, and evaluation protocols for our VLM and LLM experiments. We then present the main results, where we compare SELR against baselines on a suite of benchmarks and datasets. Finally, we conduct a comprehensive ablation study to validate our design choices and provide insights into the effectiveness and interpretability of our latent reasoning method.

## 4.1. Experiment Setup

### 4.1.1. DATASET

**VLM Training.** We use the LLaVA-CoT-100k dataset (Xu et al., 2025) for VLM training. This is a specialized reasoning dataset for VLMs, comprising 100k image-text pairs. It integrates samples from several widely used VQA datasets and is notable for providing three stages of CoT reasoning for each sample: summary, caption, and reasoning.

**LLM Training.** For LLM experiments, we use the GSM8k-Aug dataset proposed by Deng et al. (2023). It is an augmented version of the GSM8k (Cobbe et al., 2021) training set, containing 385k grade-school math problems generated by prompting GPT-4.

### 4.1.2. MODEL TRAINING

We use Qwen2.5-VL-3B-Instruct (Bai et al., 2025) for VLM experiments and LLaMA-3.2-1B-Instruct (Meta, 2024a) for LLM experiments. All models are fine-tuned using LoRA (Hu et al., 2022) and trained with the AdamW (Loshchilov & Hutter, 2019) optimizer using the Hugging Face Accelerate library (Gugger et al., 2022). For VLM training, all SELR methods are trained for a total of 3 epochs. For LLM training, the single-stage method is trained for 10 epochs, while the multi-stage method is trained for 10 epochs in the first phase and 5 epochs in the second. Detailed hyperparameters, including LoRA config-

urations and scheduler details, are provided in Appendix A.

### 4.1.3. BASELINES

For our LLM evaluation, we compare against three methods that share the same LLaMA-3.2-1B-Instruct backbone: (1) **CoT-SFT:** Model finetuned directly on the ground-truth CoTs and answers in the GSM8k-Aug dataset for 15 epochs. (2) **Coconut:** A latent space reasoning method trained by gradually replacing textual CoT steps with latent thoughts (Hao et al., 2025). To ensure fair comparison, we retrain the method for 15 epochs as well with its original code. Specific training configurations are detailed in Appendix A. (3) **CoLaR:** A latent reasoning framework that dynamically compresses multiple consecutive reasoning tokens into single latent embeddings via an auxiliary prediction objective (Tan et al., 2025). We present its best reported results, which use a test-time compression factor of 2, for comparison.

For the VLM experiments, we utilize Qwen2.5-VL-3B-Instruct as the backbone. In addition to the base model, we compare against a LLaVA-CoT-100k fine-tuned version and a reimplementation of Heima (Shen et al., 2025a). Further implementation details are provided in Appendix A.

Here we also clarify why two other recent latent reasoning methods are not included as direct baselines. HRPO (Yue et al., 2025) mixes latent embeddings with discrete text tokens via a learnable gating mechanism rather than fully replacing textual CoT with latent representations, and therefore does not achieve the same reasoning length compression as purely latent methods. CODI (Shen et al., 2025b) compresses CoT into continuous space via self-distillation but does not provide any mechanism for decoding or explaining its latent representations. Since SELR's primary contribution is self-explainability, comparing against a method without this capability would not meaningfully evaluate our core claim. Both methods are complementary to SELR rather than direct competitors.

### 4.1.4. EVALUATION

To comprehensively evaluate our methods, we test our models on a wide array of benchmarks and datasets.

**VLM Evaluation.** In accordance with Heima, we perform zero-shot evaluation on MMStar (Chen et al., 2024), MM-BenchV1.1 (Liu et al., 2024), MMVet (Yu et al., 2024), MathVista (Lu et al., 2024), AI2D (Hiippala et al., 2020), and HallusionBench (Guan et al., 2024). MMStar, MM-Bench, and MMVet are used to evaluate core visual question-answering and reasoning abilities. MathVista and AI2D test complex mathematical and scientific diagram understanding. Hallusion is used to measure the model's tendency to produce factual inaccuracies or hallucinations. The evaluations

*Table 1.* Main VLM benchmark results. All presented methods are based on Qwen2.5-VL-3B-Instruct. The value enclosed by parentheses in the Average column reflects the change from the base model. SELR not only improves base model accuracy (in contrast, Heima degrades accuracy), but also drastically reduces token count. Best results on each benchmark is shown in **bold**.

| Dataset | MMStar | | MMBench | | MMVet | | MathVista | | AI2D | | Hallusion | | Average | |
|---|---|---|---|---|---|---|---|---|---|---|---|---|---|---|
| Model | Acc. | # Token | Acc. | # Token | Acc. | # Token | Acc. | # Token | Acc. | # Token | Acc. | # Token | Acc.↑ | # Token↓ |
| Qwen2.5-VL-3B-Instruct | 54.27 | 11.92 | 76.70 | 8.60 | 44.36 | 139.19 | 62.40 | 93.83 | 78.01 | 5.25 | 61.41 | 40.72 | 62.86 | 49.92 |
| Qwen2.5-VL-3B-Instruct-SFT | 55.47 | 210.19 | 72.37 | 180.23 | **44.82** | 249.07 | 56.40 | 253.89 | 78.30 | 208.37 | 60.15 | 210.91 | 61.25 | 218.78 |
| Heima | 53.60 | 12.62 | 75.93 | 12.22 | 34.26 | 73.60 | 59.90 | 12.99 | 79.37 | 12.72 | 62.57 | 12.86 | 60.94 (-1.92) | 22.84 (-54.25%) |
| SELR (Answer Loss Only) | 56.27 | 7.27 | 76.63 | 7.12 | 36.38 | 54.65 | 61.50 | 9.60 | 78.30 | 7.56 | 59.94 | 7.28 | 61.50 (-1.36) | 15.58 (-68.79%) |
| **SELR (Single)** | **57.27** | 7.56 | 76.32 | 7.29 | 41.88 | 56.83 | **65.10** | 7.51 | 78.95 | 8.00 | 62.78 | 7.12 | **63.72** (+0.86) | 15.72 (-68.51%) |
| **SELR (Multi, Uniform)** | **57.27** | 7.67 | 77.40 | 7.32 | 38.99 | 40.75 | 64.30 | 7.52 | 79.89 | 8.56 | 62.99 | 8.07 | 63.47 (+0.61) | 13.32 (-73.32%) |
| **SELR (Multi, Exponential)** | 56.40 | 7.70 | **78.72** | 7.37 | 36.10 | 42.76 | 64.30 | 7.52 | **80.31** | 8.15 | **64.04** | 7.34 | 63.31 (+0.45) | 13.47 (-73.02%) |

*Table 2.* LLM results on GSM8k, SVAMP, GSM-Hard, and MultiArith. All models are based on LLaMA-3.2-1B-Instruct. SELR (Multi) consistently improves upon the single-stage variant and the Coconut baseline, and remains highly competitive with CoLaR.

| Model | GSM8k | SVAMP | GSM-Hard | MultiArith |
|---|---|---|---|---|
| Coconut (Hao et al., 2025) | 30.83 | 36.33 | 0.00 | 80.00 |
| CoLaR (Tan et al., 2025) | 40.1 | 54.9 | 9.08 | 91.3 |
| CoT-SFT | 64.06 | 66.67 | 15.85 | 98.33 |
| **SELR (Single)** | 35.03 | 46.33 | 8.04 | 70.00 |
| **SELR (Multi)** | 42.46 | 49.67 | 9.78 | 81.67 |

are conducted with VLMEvalKit (Duan et al., 2024) for standardized evaluation, using GPT-4o (OpenAI, 2024a) to score the responses on MMVet and MathVista, and exact matching for all other benchmarks. For decoding quality evaluation, we split the LLaVA-CoT-100k dataset into a 95% train split and a 5% test split, and retrain the SELR methods on the train split. We also utilize BLEU-4 (Papineni et al., 2002), METEOR (Banerjee & Lavie, 2005), ROUGE (Lin, 2004), BERTScore (Zhang et al., 2020), and GPT-4o to evaluate the similarity of the decoded results on the test set. The prompt used for similarity evaluation with GPT-4o is provided in Appendix F.

**LLM Evaluation.** We perform in-domain evaluation on the GSM8k test set and out-of-distribution (OOD) evaluation on SVAMP (Patel et al., 2021), MultiArith (Roy & Roth, 2015), and GSM-Hard (Gao et al., 2023). We report the final answer accuracy for all benchmarks.

### 4.2. Main Results

#### 4.2.1. VLM RESULTS

In Table 1, Qwen2.5-VL-3B-Instruct-SFT is the base model finetuned on LLaVA-CoT-100k for 3 epochs; SELR (Answer Loss Only) refers to the method of training the model with only the Answer Loss; SELR (Single) refers to the single-stage method for VLM training (Section 3.3.1); SELR (Multi, Uniform) and SELR (Multi, Exponential) refer to the multi-stage method for VLM training (Section 3.3.2), using the uniform distribution and exponential distribution for the Single Step Loss, respectively.

In terms of performance, Table 1 shows that SELR has a consistent positive gain over the average performance of the original model, improving by 0.86% at best. As comparison, Heima's performance is worse than the base model, showing a 1.92% drop. This shows that our method, despite introducing great changes in the model's generation paradigm, still preserves and even improves the model's general and reasoning capabilities.

SELR also shows great improvements on efficiency, reducing the response length by a large margin, while improving performance at the same time. Our multi-stage variants achieve an average response length of approximately 13 tokens and a reduction of over 70% from our base model. Our model responses are also more concise than the Heima baseline, which averages 22.84 tokens per response. We further validate generalization at the 7B scale in Appendix H and on a different training dataset without fixed-format structure in Appendix J. We also report wall-clock inference latency in Appendix G, where SELR achieves an 8.7× speedup over SFT.

**MMVet Performance.** The MMVet score for SELR multi-stage variants is lower than the base model. This is not a general degradation but is localized to specific capability splits: the decline is driven by compromised multi-step reasoning (*e.g.*, math sub-problems) and long-form generation capabilities (*e.g.*, free-form writing), both of which rely on extended token generation that latent compression naturally limits. This is a shared challenge for latent-space thinking models: Heima also degrades on MMVet, and more severely than SELR (34.26 vs. 36.10). SELR outperforms baselines on pure visual perception tasks within MMVet (*e.g.*, direct OCR, spatial localization), partially offsetting the overall decline.

**SFT Underperformance.** Qwen2.5-VL-3B-Instruct-SFT performs worse on average than the original model despite being fine-tuned on LLaVA-CoT-100k. This is not a bug but an expected consequence of training on the specially structured CoT data: the dataset requires responses in a rigid three-step format, introducing a format shift from the base model's default behavior. This forces the model to produce multi-stage reasoning even for simple questions

where the base model would answer concisely, and the structured format may conflict with pre-training reasoning patterns. We observe the same degradation at the 7B scale in Table 16.

**Token Count.** The token counts in Table 1 are the total generated output tokens, including latent special tokens `<bot>`/`<eot>` and the final answer, *not* decoded reasoning tokens.

### 4.2.2. LLM RESULTS

The LLM results are presented in Table 2. In the table, SELR (Single) refers to the model trained with the single-stage method (Section 3.3.1); SELR (Multi) refers to the model trained with the multi-stage method (Section 3.3.2).

As we can see from Table 2, SELR (Multi) outperforms the Coconut baseline and our single-stage variant on both in-domain (*e.g.,*, GSM8k) and out-of-domain (OOD) datasets (*e.g.,*, SVAMP), demonstrating the effectiveness of the multi-stage training curriculum. Furthermore, it remains highly competitive with CoLaR, notably surpassing it on the in-domain GSM8k dataset and the challenging GSM-Hard benchmark. It is also notable that the reasoning length of CoLaR is longer than SELR, being at most more than twice as long (14.0 vs. 6.0).

We note that the remaining gap between SELR and CoT-SFT reflects a difference in reasoning budget rather than reasoning quality: CoT-SFT generates over 150 text tokens per question, while SELR uses only 6 latent tokens. When CoT-SFT is constrained to the same 6-token budget, its performance collapses to 10.31% on GSM8k, compared to SELR's 42.46% . See Appendix I for the full controlled-budget comparison.

### 4.2.3. DECODING QUALITY

As mentioned in Section 4.1.4, we also evaluate the decoding results with several evaluation metrics and GPT-4o. The results are presented in Table 3.

As we can see from Table 3, the summaries, captions, and reasoning decoded from the latent thoughts are more similar to the ground truth than Heima. The VLM methods that use the Single Step Loss can decode the corresponding latent thoughts back to textual summaries, captions, and reasoning consistently better. Notably, in addition to the Heima encoder, Heima trains a decoder for each of the latent thoughts (summary, caption, reasoning), therefore 4 models are involved to decode the latent thoughts. However, in SELR, only one model is needed. Additionally, the Heima decoders are based on LLaMA3.1-8B-Instruct (Meta, 2024a), which is more than twice the size of our backbone. Despite this significant architectural advantage favoring the Heima baseline, our method achieves superior decoding quality. This

*Table 3.* Decoding quality evaluation. We compare the similarity of the decoded text to the ground-truth for the **Summary**, **Caption**, and **Reasoning** steps from the LLaVA-CoT-100k test split. Our SELR methods consistently outperform the Heima baseline across all steps.

| Model | BLEU-4 | METEOR | ROUGE-L | BERTScore | GPT-4o |
|---|---|---|---|---|---|
| *Summary* | | | | | |
| Heima | 15.9 | 40.1 | 41.6 | 73.4 | 4.1 |
| SELR (Multi, Exponential) | 19.00 | 44.25 | 43.72 | 76.78 | 4.24 |
| SELR (Multi, Uniform) | **19.95** | **45.05** | **44.60** | **77.19** | **4.30** |
| *Caption* | | | | | |
| Heima | 12.8 | 35.5 | 37.9 | 71.4 | 2.7 |
| SELR (Multi, Exponential) | 18.31 | 43.22 | 42.74 | 75.04 | 3.26 |
| SELR (Multi, Uniform) | **18.42** | **43.27** | **43.01** | **75.09** | **3.27** |
| *Reasoning* | | | | | |
| Heima | 11.2 | 32.7 | 32.7 | 66.6 | 3.2 |
| SELR (Multi, Exponential) | **11.37** | **34.28** | **32.08** | **66.83** | **3.42** |
| SELR (Multi, Uniform) | 10.86 | 33.56 | 31.59 | 66.54 | 3.38 |

*Table 4.* Full CoT Loss vs. Single Step Loss for decoding. We compare the full-sequence decoding quality of the single-stage method against the multi-stage methods. For the multi-stage methods, the full CoT is acquired by concatenating the decoded summary, caption, and reasoning.

| Model | BLEU-4 | METEOR | ROUGE-L | BERTScore | GPT-4o |
|---|---|---|---|---|---|
| SELR (Single) | 19.72 | 38.88 | 38.04 | **70.01** | 3.25 |
| SELR (Multi, Exponential) | **20.10** | **39.32** | 38.32 | 69.92 | **3.46** |
| SELR (Multi, Uniform) | 20.07 | 38.95 | **38.37** | 69.86 | 3.44 |

observation demonstrates the synergy enabled by unifying reasoning and self-explanation in a shared model.

When evaluating CoT similarity of the full decoded sequence with the original CoT sequence in Table 4, it is evident that the VLM methods that use the Single Step Loss achieve better similarity, which confirms our intuition that step-wise decoding is better than decoding the full sequence at the same time for VLMs.

### 4.2.4. FAITHFULNESS ANALYSIS

A natural question is whether the decoded CoT faithfully reflects the model's actual reasoning process. We address this with a *consistency metric*: GPT-4o scores the alignment between the decoded thoughts and the model's final answer on a 0–1 scale, evaluated on a held-out 5% test split of LLaVA-CoT-100k. Results are shown in Table 5.

The substantial gap between correct ($\sim$0.5) and incorrect ($\sim$0.26) predictions demonstrates that the decoded CoT meaningfully correlates with the model's reasoning success, suggesting it reflects genuine reasoning content rather than generic post-hoc outputs. Among variants, SELR (Multi, Exponential) yields the highest faithfulness, validating our design choices.

To directly compare against Heima, we evaluate both methods under the same consistency metric on MathVista. As shown in Table 6, SELR (Single) achieves a consistency score of 0.2097, substantially higher than Heima's 0.1671 (+25.5% relative improvement). This advantage stems from

*Table 5.* Faithfulness analysis via consistency metric. We report GPT-4o alignment scores between decoded CoT and the final answer, stratified by whether the model answered correctly.

| Method | Overall | Correct | Incorrect |
|---|---|---|---|
| SELR (Single) | 0.4481 | 0.4907 | 0.2621 |
| SELR (Multi, Uniform) | 0.4677 | 0.5118 | 0.2722 |
| SELR (Multi, Exponential) | **0.4788** | **0.5320** | **0.2399** |

*Table 6.* Consistency comparison with Heima on MathVista. SELR achieves substantially higher consistency scores, indicating that its decoded thoughts are more aligned with the model's final answer.

| Method | Consistency (MathVista) |
|---|---|
| Heima | 0.1671 |
| **SELR (Single)** | **0.2097** (+25.5% relative) |

*Table 7.* Ablation study of single-stage SELR on LLM benchmarks. Results show that the CoT Loss and the fixed latent length are critical for performance.

| Model | GSM8k | SVAMP | GSM-Hard | MultiArith |
|---|---|---|---|---|
| SELR (Single) | 35.03 | 46.33 | 8.04 | 70.00 |
| - w/ Variable Latent Length | 32.83 | 48.00 | 7.35 | 66.67 |
| - w/o Full CoT Loss | 33.97 | 40.33 | 7.73 | 64.44 |

*Table 8.* Ablation study of multi-stage SELR on LLM benchmarks. We validate the effectiveness of the components in the multi-stage method. The results confirm the benefits of using an exponential sampling distribution, including the Single Step Loss in Stage 1, and using a fixed latent length.

| Model | GSM8k | SVAMP | GSM-Hard | MultiArith |
|---|---|---|---|---|
| SELR (Multi) | 42.46 | 49.67 | 9.78 | 81.67 |
| - w/ Uniform Distribution | 37.30 | 46.00 | 8.34 | 76.11 |
| - w/o Single Step Loss (Stage 1) | 37.98 | 52.67 | 8.34 | 76.11 |
| - w/ Variable Latent Length | 38.36 | 58.33 | 8.72 | 74.44 |

SELR's unified architecture: the same model parameters responsible for reasoning are also responsible for decoding, inherently aligning the explanation with the actual computation. In contrast, Heima's separate-decoder design creates a structural gap between the model that reasons and the model that explains. Combined with the decoding-quality comparison in Table 3, this provides end-to-end evidence that SELR is more faithful than Heima on both proxies for faithfulness.

### 4.3. Ablation Study

#### 4.3.1. VLM ABLATIONS

**CoT Loss.** Comparing SELR (Answer Loss Only) with the other three SELR methods in Table 1, we can see that training the model only on the Answer Loss is detrimental to the model's performance. In stark contrast, introducing any type of CoT Loss immediately reverses this degradation, yielding gains over the base model. This strongly indicates that by forcing the latent thoughts to be explainable, it guides the model to learn more effective and structured representations, which in turn improves the quality of the answer generation. This conclusion is further corroborated with LLM ablations, detailed in the next section.

**SELR vs. Supervised Fine-Tuning.** We conduct an ablation to isolate the source of our performance gains. A key question is whether the improvement comes from our SELR framework or simply from exposing the model to the LLaVA-CoT-100k dataset. To test this, we trained a Qwen2.5-VL-3B-Instruct-SFT baseline, which fine-tunes the base model on the same dataset using a standard textual CoT objective. As shown in Table 1, this SFT approach is less performant. This demonstrates that the performance gain is not an artifact of the dataset, but is a direct result of our methods. The SELR methods successfully teaches the model to internalize the reasoning process, leading to higher

accuracy and reduction in response length.

#### 4.3.2. LLM ABLATIONS

**Single-Stage Methods.** Similar to the VLM ablations, Table 7 shows that the CoT loss is beneficial to the answer generation. Also, we can see that training with a fixed latent length is better than training with variable length, which coincides with Coconut's observation that training an auxiliary model to predict the ending of the latent space reasoning achieves similar results to fixing the latent space reasoning length.

**Multi-Stage Methods.** As in Table 8, each option is integral to the final performance of the multi-stage method: the exponential distribution focuses on the more difficult ending steps; the Answer Loss during the first phase helps with regularizing the latent thoughts; and again, using a fixed latent length helps with performance.

## 5. Conclusion

In this work, we propose Self-Explainable Latent Reasoning (SELR), a unified framework that trains a single, unified model to be both an efficient reasoner and its own translator. Our core contribution is a multi-task learning objective that combines a standard answer loss with a CoT loss. Experimental results demonstrate that our method is effective for both VLM and LLM tasks, yielding performance gains while significantly reducing response length. By successfully bridging the gap between efficiency and interpretability, SELR provides a practical path toward developing efficient and explainable latent space reasoning models. A promising avenue for future work is the development of a stable, learned stopping mechanism, which we have identified as a significant challenge.

## Acknowledgements

This work was supported in part by NSF under Grants 2106825 and 2519216, the DARPA Young Faculty Award, the ONR Grant N00014-26-1-2099, the NIFA Award 2020-67021-32799, the Amazon-Illinois Center on AI for Interactive Conversational Experiences, the Capital One Illinois Center for Generative AI Safety, Knowledge Systems, and Cybersecurity, the IBM-Illinois Discovery Accelerator Institute, and Apple AIML Academic Research Program. This work used computational resources, including the NCSA Delta and DeltaAI supercomputers through allocations CIS230012, CIS230013, CIS240133, and CIS240387 from the Advanced Cyberinfrastructure Coordination Ecosystem: Services & Support (ACCESS) program, as well as the TACC Frontera supercomputer, Amazon Web Services (AWS), and OpenAI API through the National Artificial Intelligence Research Resource (NAIRR) Pilot.

## Impact Statement

This paper presents work whose goal is to advance the field of Machine Learning. There are many potential societal consequences of our work, none which we feel must be specifically highlighted here.

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

# Appendix

This appendix provides comprehensive supplementary material to support the main paper. Specifically, Appendix A details our experimental setups. Appendix B presents additional qualitative examples and visualizations of decoded latent thoughts. Appendix C evaluates our Self-Explainable Latent Reasoning (SELR) framework on alternative SFT base configurations. Appendix D and Appendix E present ablation studies regarding the choice of fixed latent lengths and the evaluation protocol for variable-length architectures, respectively. Appendix F provides the exact prompt template utilized for GPT-4o automated evaluation. Appendix G reports empirical wall-clock inference latencies. Appendix H demonstrates performance scaling results on a larger 7B parameter foundation model. Appendix I provides a controlled-budget analysis contrasting SELR against Chain-of-Thought (CoT) fine-tuning. Appendix J evaluates out-of-distribution dataset generalization on unstructured text annotations. Appendix K discusses the ordering design choices for visual reasoning steps. Finally, Appendix L outlines the limitations of this work along with avenues for future research.

## A. Implementation Details

In this section, we introduce the details of our model training and evaluation.

### A.1. Training Configuration

We first detail the hyperparameters used for training our SELR methods, as well as those for the reimplemented baselines. The specific hyperparameters for experiments on LLMs are listed in Table 9, while the hyperparameters for VLM experiments are provided in Table 10.

As outlined in Section 3.3, the multi-stage SELR method on LLMs is divided into two distinct stages. The first stage consists of 10 epochs with a learning rate of 5e-4, followed by a second stage of 5 epochs with a reduced learning rate of 1e-4. This results in a total training duration of 15 epochs.

For Coconut (Hao et al., 2025), to ensure a rigorous comparison, we reproduce it directly using the official repository code[1]. We set the total training budget to 15 epochs to match the setting of our method: following the original paper's curriculum setup, the model is trained for 3 epochs in the initial stage, followed by 3 epochs for each of the remaining stages (comprising 3 intermediate stages and 1 final stage). It is important to note that the original Coconut implementation employs full fine-tuning with full precision. Consequently, the computational cost for the original training setup for

Coconut is significantly higher, requiring over 50 hours on four NVIDIA A100 GPUs, while our method requires about 30 hours.

The Qwen2.5-VL-3B-Instruct-SFT model is Qwen2.5-VL-3B-Instruct (Bai et al., 2025) finetuned on LLaVA-CoT-100k (Xu et al., 2025). More specifically, it is finetuned on LLaVA-CoT-100k with the structured XML tags (*e.g.*, <SUMMARY> and </SUMMARY>) included. In the original paper, it is mentioned that training without the structured XML tags can cause degradation in performance, and it is recommended to finetune with the tags included. We also provide another version, Qwen2.5-VL-3B-Instruct-SFT (w/o Structured Tags), in Table 11, which is finetuned on LLaVA-CoT-100k without the tags. The results agree with the LLaVA-CoT paper, showing a 5.14% drop in average performance if the base model is finetuned without tags.

For Heima (Shen et al., 2025a), we adhere to the training configurations and curriculum provided in the original paper. Since the original model is based on LLaVA-CoT, which is essentially Llama-3.2-11B-Vision-Instruct (Meta, 2024b) fine-tuned on LLaVA-CoT-100k, our reimplementation of Heima is trained on Qwen2.5-VL-3B-Instruct-SFT. In the progressive decoding phase, we gradually transform the reasoning steps into three corresponding thinking tokens, dedicating one epoch to the distillation of each. Subsequently, we train for an additional epoch in the recovering phase using a slightly lower learning rate, which effectively repeats the final stage of the progressive encoding phase.

### A.2. Inference Settings

We use greedy decoding for LLM inference and sampling-based decoding for VLM inference. For sampling, we set the temperature to 1e-2, top-$p$ to 1e-3, and set a repetition penalty of 1.0, which are essentially the default values provided for Qwen2.5-VL-3B-Instruct (Bai et al., 2025) in VLMEvalKit (Duan et al., 2024), so that we can ensure a fair comparison with baselines.

## B. Additional Qualitative Results

In this section, we present additional qualitative examples to further demonstrate the effectiveness of our SELR method compared to baselines. As shown in Figure 5, our method gives concise responses, improving token efficiency significantly. Furthermore, our method shows better performance over the base model in these examples.

We also provide additional examples in Figure 6, demonstrating that our method produces human-readable and coherent reasoning that leads to the correct answers.

---

[1]https : / / github . com / facebookresearch / coconut

*Table 9.* **Training Hyperparameters for LLM Experiments.** N/A means that the corresponding hyperparameter is not applicable to the experiment. For example, all LoRA hyperparameters are N/A for Coconut, because it performs full fine-tuning without LoRA.

| Experiment | SELR (Single) | SELR (Multi) | Coconut | CoT-SFT |
|---|---|---|---|---|
| Epoch | 10 | 15 | 15 | 3 |
| Batch Size | 128 | 128 | 128 | 128 |
| LoRA Rank | 128 | 128 | N/A | 128 |
| LoRA Alpha | 32 | 32 | N/A | 32 |
| LoRA Dropout | 0.1 | 0.1 | N/A | 0.1 |
| Optimizer | AdamW | AdamW | AdamW | AdamW |
| Weight Decay | 0.1 | 0.1 | 0.01 | 0.1 |
| Learning Rate | 5e-04 | 5e-04, 1e-04 | 1e-04 | 5e-04 |
| Scheduler | Cosine | Cosine | N/A | Cosine |
| Warmup Ratio | 0.1 | 0.1 | N/A | 0.1 |
| Max Gradient Norm | 1.0 | 1.0 | N/A | 1.0 |
| DeepSpeed | N/A | N/A | N/A | N/A |
| Mixed Precision (bf16) | True | True | False | True |

*Table 10.* **Training Hyperparameters for VLM Experiments.** The SFT baseline and SELR variants are fine-tuned on Qwen2.5-VL-3B-Instruct for 3 epochs with identical optimization settings to ensure a fair comparison. Heima is trained in accordance with its proposed curriculum.

| Experiment | SELR | Qwen2.5-VL-3B-Instruct-SFT | Heima |
|---|---|---|---|
| Epoch | 3 | 3 | 4 |
| Batch Size | 8 | 8 | 8 |
| LoRA Rank | 128 | 128 | 128 |
| LoRA Alpha | 32 | 32 | 32 |
| LoRA Dropout | 0.1 | 0.1 | 0.1 |
| Optimizer | AdamW | AdamW | AdamW |
| Weight Decay | 0.1 | 0.1 | 0.1 |
| Learning Rate | 5e-04 | 5e-05 | 5e-04, 5e-05 |
| Scheduler | Cosine | Cosine | Cosine |
| Warmup Ratio | 0.1 | 0.1 | 0.1 |
| Max Gradient Norm | 1.0 | 1.0 | 1.0 |
| DeepSpeed | ZeRO-2 | N/A | N/A |
| Mixed Precision (bf16) | True | True | True |

## C. SELR on Qwen2.5-VL-3B-Instruct-SFT

For a fair comparison, it is necessary to isolate the performance gains attributed to our latent reasoning framework from those obtained simply by fine-tuning on the target dataset. In Table 11, we present the results of SELR (Single), SELR (Multi, Uniform), and SELR (Multi, Exponential) when they are trained based on Qwen2.5-VL-3B-Instruct-SFT (w/o Structured Tags). The training hyperparameters are identical to those listed in Table 10. As we can see from Table 11, our method still shows consistent gains in terms of performance and efficiency, with an approximately 6% gain over Qwen2.5-VL-3B-Instruct-SFT (w/o Structured Tags) in performance, and over 90% gain in token efficiency.

Although Qwen2.5-VL-3B-Instruct-SFT (w/o Structured Tags) has a poorer performance relative to Qwen2.5-VL-3B-Instruct-SFT, we train on this version nevertheless to avoid the shifting of generation pattern (*i.e.*, with or without the tags). Despite this less performant base model, the models trained with SELR still surpass Qwen2.5-VL-3B-Instruct-SFT, underscoring the efficacy of our method.

**Example: Unfaithful CoT Decoding**

```
Question:  "Tom's ship can travel
at 10 miles per hour.  He is
sailing from 1 to 4 PM. He then
travels back at a rate of 6 mph.
How long does it take him to get
back?"
Decoded CoT = "<<4-1=3>>
<<3*10=30>> <<30/6=5>>"
Direct Answer = "2"
```

*Figure 4.* **An example of unfaithful CoT decoding.** The decoded CoT is correct, but the direct answer is wrong, indicating that the decoded CoT is not the true reasoning path for the model's answer.

## D. Latent Length Ablation

For the LLM experiments, we use a latent length of 6 to maintain consistency with the Coconut baseline. To validate this hyperparameter choice, we conduct an ablation study on the latent sequence length. Note that this is not viable for the SELR method on VLMs, because Single Step Loss requires one-to-one correspondence between the latent thoughts and the reasoning steps.

The results are shown in Table 12. For these experiments, we stick to the training hyperparameters listed in the SELR (Single) configuration of Table 9, except for the training duration, which is reduced to 2 epochs for efficiency. The results indicate that a latent length of 6 provides the optimal balance; reducing the length to 2 or 4 limits the model's reasoning capacity, while increasing it to 8 does not yield significant improvements.

## E. Evaluating Variable Latent Length Methods

We now clarify the evaluation protocol for our SELR method trained with variable latent length in Tables 7 and 8. As discussed in Section 3.2.1, it is hard to train the model to predict the <eot> token, so we have to cut off the latent reasoning generation with a fixed budget during evaluation. Specifically, an inference budget of $b$ implies that the model generates exactly $b$ latent thought tokens following the <bot> token. We then terminate the latent reasoning generation process, append the <eot> token, and proceed to generate the final answer.

In Tables 7 and 8, we report the optimal accuracy achieved across all tested budgets for each benchmark. This provides an upper-bound performance estimate for the variable length method. Notably, even when granted this "oracle" budget selection, the variable length approach generally underperforms compared to our proposed fixed latent length strategy.

*Table 11.* **Performance of SELR variants trained on Qwen2.5-VL-3B-Instruct-SFT.** We report the accuracy and average token count per response across six benchmarks. Qwen2.5-VL-3B-Instruct-SFT and Qwen2.5-VL-3B-Instruct-SFT (w/o Structured Tags) are Qwen2.5-VL-3B-Instruct finetuned on LLaVA-CoT-100k with and without structured tags, respectively. The value enclosed by the parentheses in the Average column reflects the change from the Qwen2.5-VL-3B-Instruct-SFT model. SELR consistently outperforms both SFT baselines in average accuracy while reducing token usage by over 90%.

| Dataset | MMStar | | MMBench | | MMVet | | MathVista | | AI2D | | Hallusion | | Average | |
|---|---|---|---|---|---|---|---|---|---|---|---|---|---|---|
| Model | Acc. | # Token | Acc. | # Token | Acc. | # Token | Acc. | # Token | Acc. | # Token | Acc. | # Token | Acc. | # Token |
| Qwen2.5-VL-3B-Instruct-SFT | 55.47 | 210.19 | 72.37 | 180.23 | 44.82 | 249.07 | 56.40 | 253.89 | 78.30 | 208.37 | 60.15 | 210.91 | 61.25 | 218.78 |
| Qwen2.5-VL-3B-Instruct-SFT (w/o Structured Tags) | 48.40 | 185.36 | 62.00 | 149.75 | 42.98 | 207.69 | 58.90 | 229.81 | 62.63 | 179.12 | 61.72 | 180.23 | 56.11 | 188.66 |
| **SELR (Single)** | 57.27 | 7.60 | 77.94 | 7.32 | 34.40 | 52.01 | 63.00 | 7.48 | 79.92 | 7.58 | 62.36 | 7.21 | 62.48 (+1.23) | 14.87 (-93.20%) |
| **SELR (Multi, Uniform)** | 56.33 | 7.40 | 76.70 | 7.26 | 38.62 | 41.91 | 62.90 | 7.54 | 79.53 | 7.47 | 61.41 | 7.26 | 62.58 (+1.33) | 13.14 (-93.99%) |
| **SELR (Multi, Exponential)** | 54.66 | 7.40 | 76.70 | 7.31 | 40.41 | 45.72 | 63.60 | 7.48 | 79.79 | 7.35 | 62.88 | 7.46 | 63.01 (+1.76) | 13.79 (-93.70%) |

*Table 12.* **Ablation study on latent length for LLM benchmarks.** We train the SELR (Single) method for 2 epochs across varying latent lengths. The results confirm that $l_{latent} = 6$ achieves the best performance across the majority of datasets.

| Latent Length | 2 | 4 | 6 | 8 |
|---|---|---|---|---|
| GSM8k | 18.12 | 22.74 | **31.69** | 31.54 |
| SVAMP | 36.33 | 39.67 | **47.67** | 47.33 |
| GSM-Hard | 4.40 | 5.84 | **7.96** | 7.20 |
| MultiArith | 36.67 | 48.33 | 61.11 | **70.00** |

For completeness, we provide the full evaluation results with different budgets in Tables 13 and 14.

*Table 13.* **Evaluation results for the single-stage SELR method trained with variable latent length.** We evaluate the model with fixed budgets ranging from 1 to 5 latent tokens. The optimal result for each dataset is in **bold**.

| Budget | 1 | 2 | 3 | 4 | 5 |
|---|---|---|---|---|---|
| GSM8k | 27.07 | **32.83** | 31.99 | 28.53 | 24.03 |
| SVAMP | **48.00** | 29.00 | 25.67 | 19.67 | 17.00 |
| GSM-Hard | 6.07 | 7.20 | **7.35** | 6.82 | 5.84 |
| MultiArith | 44.44 | **66.67** | 58.89 | 38.89 | 25.56 |

## F. Prompt for GPT-4o Evaluation

As mentioned in Section 4.1.4, we utilize GPT-4o to evaluate the similarity between the decoded CoTs and the ground truth CoTs. To ensure a fair comparison with Heima (Shen et al., 2025a), we use the identical prompt that Heima uses, listed in the original paper's appendix. Here, we provide a copy of it for clarity in Algorithm 2.

## G. Inference Latency

To confirm that SELR's token reduction translates into real wall-clock efficiency gains, we report per-sample inference latency on MathVista in Table 15. SELR (Single) achieves the highest accuracy (65.10%) and the lowest latency (0.81s per sample), yielding an **8.7× speedup** over the SFT baseline.

*Table 14.* **Evaluation results for the multi-stage SELR method trained with variable latent length.** We evaluate the model with fixed budgets ranging from 1 to 5 latent tokens. The optimal result for each dataset is in **bold**.

| Budget | 1 | 2 | 3 | 4 | 5 |
|---|---|---|---|---|---|
| GSM8k | 30.40 | 36.62 | **38.36** | 34.19 | 30.10 |
| SVAMP | **58.33** | 39.00 | 32.00 | 29.33 | 26.33 |
| GSM-Hard | 6.44 | **8.72** | 8.57 | 7.81 | 6.44 |
| MultiArith | 61.11 | **74.44** | 65.56 | 56.11 | 52.78 |

*Table 15.* Per-sample inference latency on MathVista. SELR achieves the best accuracy with the lowest latency, demonstrating that token reduction directly translates into wall-clock speedup.

| Method | MathVista Acc. | Latency (s/sample) | Speedup |
|---|---|---|---|
| Qwen2.5-VL-3B-Instruct-SFT | 56.40 | 7.05 | 1.0× |
| Qwen2.5-VL-3B-Instruct | 62.40 | 1.64 | 4.3× |
| **SELR (Single)** | **65.10** | **0.81** | **8.7×** |

## H. Qwen2.5-VL-7B-Instruct Experiments

To validate generalization across model scales, we trained SELR on Qwen2.5-VL-7B-Instruct (Bai et al., 2025). Table 16 presents the results. The same pattern holds at the 7B scale: SELR improves average accuracy over the original model (+0.11), SFT degrades accuracy despite generating far more tokens, and SELR achieves over 80% token reduction. This confirms that SELR's benefits generalize across model scales.

## I. Controlled-Budget LLM Comparison

A key question is whether the performance gap between SELR and CoT-SFT on LLM benchmarks reflects a difference in reasoning mechanism or simply in reasoning budget. CoT-SFT generates over 150 reasoning tokens per question on GSM8k, while SELR uses only 6 latent tokens. To disentangle this, we constrain CoT-SFT to a comparable budget by truncating its generation to 6 tokens plus the answer, and also report its direct-answer (zero reasoning token) performance. Results are shown in Table 17.

The results clearly demonstrate that CoT-SFT's advantage

*Table 16.* **Results on Qwen2.5-VL-7B-Instruct.** Each cell shows Accuracy (Avg. #Tokens). SELR improves accuracy over the original model while achieving over 80% token reduction, confirming generalization across model scales.

| Dataset | MMStar | | MMBench | | MMVet | | MathVista | | AI2D | | Hallusion | | Average | |
|---|---|---|---|---|---|---|---|---|---|---|---|---|---|---|
| Model | Acc. | # Token | Acc. | # Token | Acc. | # Token | Acc. | # Token | Acc. | # Token | Acc. | # Token | Acc. | # Token |
| Qwen2.5-VL-7B-Instruct | 61.73 | 55.0 | 82.28 | 13.3 | **46.38** | 138.1 | 68.10 | 202.8 | 81.15 | 5.6 | 65.62 | 72.6 | 67.54 | 81.2 |
| Qwen2.5-VL-7B-Instruct-SFT | 62.53 | 213.7 | 78.17 | 184.2 | 44.08 | 252.9 | 65.10 | 257.3 | 82.12 | 203.7 | 62.78 | 214.4 | 65.80 | 221.0 |
| **SELR (Single)** | **62.93** | 8.6 | **83.05** | 8.9 | 43.35 | 33.3 | **68.60** | 7.4 | **82.16** | 9.7 | **65.83** | 8.4 | **67.65** | 12.7 |

*Table 17.* **Controlled-budget LLM comparison.** When CoT-SFT is constrained to the same 6-token budget as SELR, its performance collapses. SELR achieves 42.46% on GSM8k vs. CoT-SFT's 10.31% under the same budget, demonstrating far more efficient reasoning encoding.

| Method | GSM8k | GSM-Hard | SVAMP | MultiArith |
|---|---|---|---|---|
| CoT-SFT (Full Budget) | 64.06 | 15.85 | 66.67 | 98.33 |
| CoT-SFT (6 Tokens + Answer) | 10.31 | 2.43 | 50.67 | 50.00 |
| CoT-SFT (Direct Answer) | 6.29 | 1.90 | 42.00 | 5.56 |
| Coconut (Hao et al., 2025) | 30.83 | 0.00 | 36.33 | 80.00 |
| CoLaR (Tan et al., 2025) | 40.1 | 9.08 | 54.9 | 91.3 |
| **SELR (Multi)** | **42.46** | **9.78** | 49.67 | 81.67 |

*Table 18.* **Results on VisualWebInstruct-Verified.** SELR outperforms CoT-SFT on 5 of 6 benchmarks, confirming that SELR's mechanism is CoT-format agnostic and generalizes to unstructured reasoning datasets.

| Method | MMStar | MMBench | MMVet | MathVista | AI2D | Hallusion | Avg |
|---|---|---|---|---|---|---|---|
| CoT-SFT | 42.46 | 40.71 | **43.03** | 56.40 | 49.90 | 58.04 | 48.42 |
| **SELR** | **52.00** | **73.53** | 37.84 | **58.20** | **77.75** | **59.73** | **59.84** |

comes from its much longer reasoning trace, not from a superior reasoning mechanism. When constrained to the same 6-token budget, CoT-SFT achieves only 10.31% on GSM8k, compared to SELR's 42.46%—a 4× advantage for SELR. This confirms that SELR encodes reasoning information far more efficiently than textual CoT on a per-token basis.

## J. Generalization to VisualWebInstruct-Verified

To verify that SELR's effectiveness is not limited to the LLaVA-CoT-100k dataset or its specific structured format, we train both SELR and CoT-SFT on VisualWebInstruct-Verified (Jia et al., 2025), a more recent VLM reasoning dataset with a completely different source and format. Results on Qwen2.5-VL-3B-Instruct are shown in Table 18.

This demonstrates that SELR works with both structured and unstructured reasoning annotations, and the improvements are not an artifact of a particular dataset structure.

## K. Discussion on Reasoning Step Ordering

For the VLM setting, SELR uses a fixed ordering of reasoning steps: summary → caption → reasoning. This ordering is inherited from the LLaVA-CoT-100k dataset (Xu et al.,

2025) and is also adopted by LLaVA-CoT and Heima (Shen et al., 2025a).

This fixed ordering offers several advantages:

1. **Natural reasoning flow.** The ordering reflects a natural reasoning process for visual question-answering: first understand the question (summary), then describe relevant visual content (caption), and finally reason toward the answer. Each step logically builds upon the previous one.

2. **Positional specialization.** A fixed order allows each latent position to specialize (*e.g.*, the first latent always encodes summaries), simplifying learning. Randomizing the order would require the model to additionally learn to identify the type of information in each position, increasing the learning difficulty.

3. **Consistency with prior work.** The same ordering is used by all prior methods trained on LLaVA-CoT-100k, ensuring fair comparison.

We note that this ordering question applies *only* to the VLM setting; in the LLM setting, SELR uses unstructured CoT steps that follow the natural reasoning sequence without imposing any predefined structure.

## L. Limitations

Despite its positive results, our method still admittedly has some limitations.

First, faithful reasoning remains a recognized open challenge even in the text-CoT literature. Prior work has shown that text-based CoT models exhibit systematic unfaithfulness: prompt-bias injection reveals that models virtually never verbalize the biases that flip their answers (Turpin et al., 2023); behavioral perturbations of the CoT show wide variation across tasks (Lanham et al., 2023); and even state-of-the-art reasoning models acknowledge influential hints less than 40% of the time (Baker et al., 2025). These measurement techniques all rely on prompt-level interventions or token-level perturbations of discrete CoT, with no direct counterpart in continuous latent space. SELR's contribution is to make latent reasoning *decodable* in the first place—a prerequisite for any faithfulness analysis on latents—while

prior latent methods (*e.g.*, Coconut, CoLaR, CODI) cannot decode their representations at all.

We note that the term "Self-Explainable" should be interpreted as self-decodable latents with meaningful faithfulness signals, rather than a guarantee of full reasoning transparency. As shown in Figure 4, the decoded CoT can occasionally produce correct reasoning while the model fails to give the correct answer directly, indicating potential misalignment between the decoded text and the model's true internal process. Nevertheless, SELR opens the door to adapting text-CoT faithfulness techniques to latent reasoning for the first time, and our consistency metric represents a concrete first step toward this goal.

Second, our methods ultimately relies on a fixed latent reasoning length, which could be suboptimal, and it is more intuitive for the model to be able to decide when to stop the latent reasoning on its own. However, the `<eot>` token is hard to predict with the LM head, as mentioned in Section 3.2.1.

We expect that developing a more advanced and learnable stopping mechanism can help the model determine the end of the latent space thinking mode, but this is beyond our current research scope.

For future work, we hope to refine our method to provide a latent space reasoning framework that provides better explainability and flexibility, while retaining efficiency and efficacy.

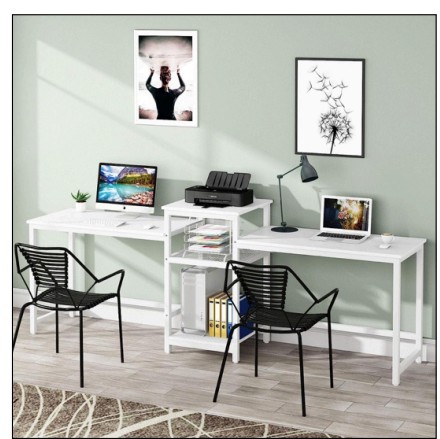

**Question:** On the right desk, what is to the left of the laptop?
**Ground Truth:** table lamp<OR>desk lamp

**LLaVA-CoT:** <SUMMARY> I will analyze the image by focusing on the right desk where the laptop is located. I will identify what is to the left of the laptop on the desk. </SUMMARY><CAPTION> The image shows a room with two white desks. The right desk has a laptop placed on it, along with a lamp and a cup. </CAPTION><REASONING> To determine what is to the left of the laptop on the right desk, I will look at the items directly beside the laptop. The image shows a cup positioned to the left of the laptop on the right desk. </REASONING><CONCLUSION> Cup </CONCLUSION> ✗

**Heima:** <CONCLUSION> Cup </CONCLUSION> ✗

**Qwen2.5-VL-3B-Instruct:** On the right desk, to the left of the laptop, there is a printer. ✗

**SELR:** Lamp. ✅

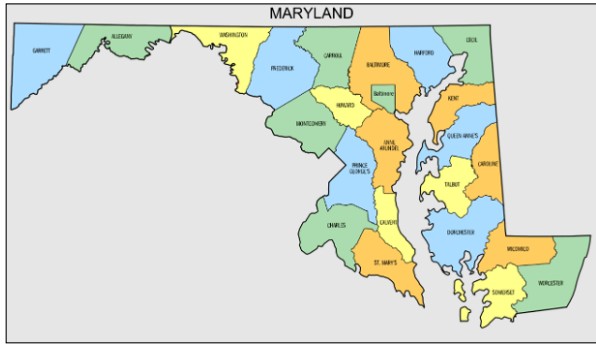

**Question:** Based on the image, in which country does Tina Fey have the highest level of total awareness?
**Ground Truth:** Australia.

**LLaVA-CoT:** <SUMMARY> I will analyze the image data to determine which country has the highest level of total awareness for Tina Fey. I will focus on the column that represents total awareness and identify the country with the highest percentage. </SUMMARY><CAPTION> The image contains two tables displaying awareness levels for two individuals. Each table is divided into sections for different countries and demographics, showing awareness percentages across various categories. The first table corresponds to a male individual, while the second table corresponds to a female individual, identified as Tina Fey. </CAPTION><REASONING> To find the country where Tina Fey has the highest total awareness, I will examine the column titled "Total" under Tina Fey's section. This column provides the overall awareness percentage for each country listed. By comparing these percentages, I can identify the country with the highest total awareness. </REASONING><CONCLUSION> United Kingdom </CONCLUSION> ✗

**Heima:** <CONCLUSION>Tina Fey has the highest level of total awareness in the UK.</CONCLUSION> ✗

**Qwen2.5-VL-3B-Instruct:** In the image, Tina Fey has the highest level of total awareness in the UK, with a total awareness of 45%. ✗

**SELR:** Australia. ✅

**Question:** On the right desk, what is to the left of the laptop?
**Ground Truth:** No, Kent County is located to the south of Cecil County.

**LLaVA-CoT:** <SUMMARY>To determine the location of Kent County in relation to Cecil County, I will analyze the map's layout and labels. I will compare the positions of both counties to assess their relative locations.</SUMMARY><CAPTION>The image is a map of Maryland, showing its counties. Each county is labeled and colored differently for clarity. The map is oriented with Maryland's coastline visible on the right </CAPTION><REASONING>First, identify the locations of Kent County and Cecil County on the map. Kent County is labeled in the northern part of the map, while Cecil County is positioned further south. By comparing their positions, it is clear that Kent County is located to the north of Cecil County.</REASONING><CONCLUSION>Yes</CONCLUSION> ✗

**Heima:** <CONCLUSION> Yes </CONCLUSION> ✗

**Qwen2.5-VL-3B-Instruct:** Yes, based on the map, Kent County is located to the north of Cecil County. ✗

**SELR:** No. ✅

*Figure 5.* More qualitative examples of LLaVA-CoT, Heima, Qwen2.5-VL-3B-Instruct, and our SELR on multimodal tasks.

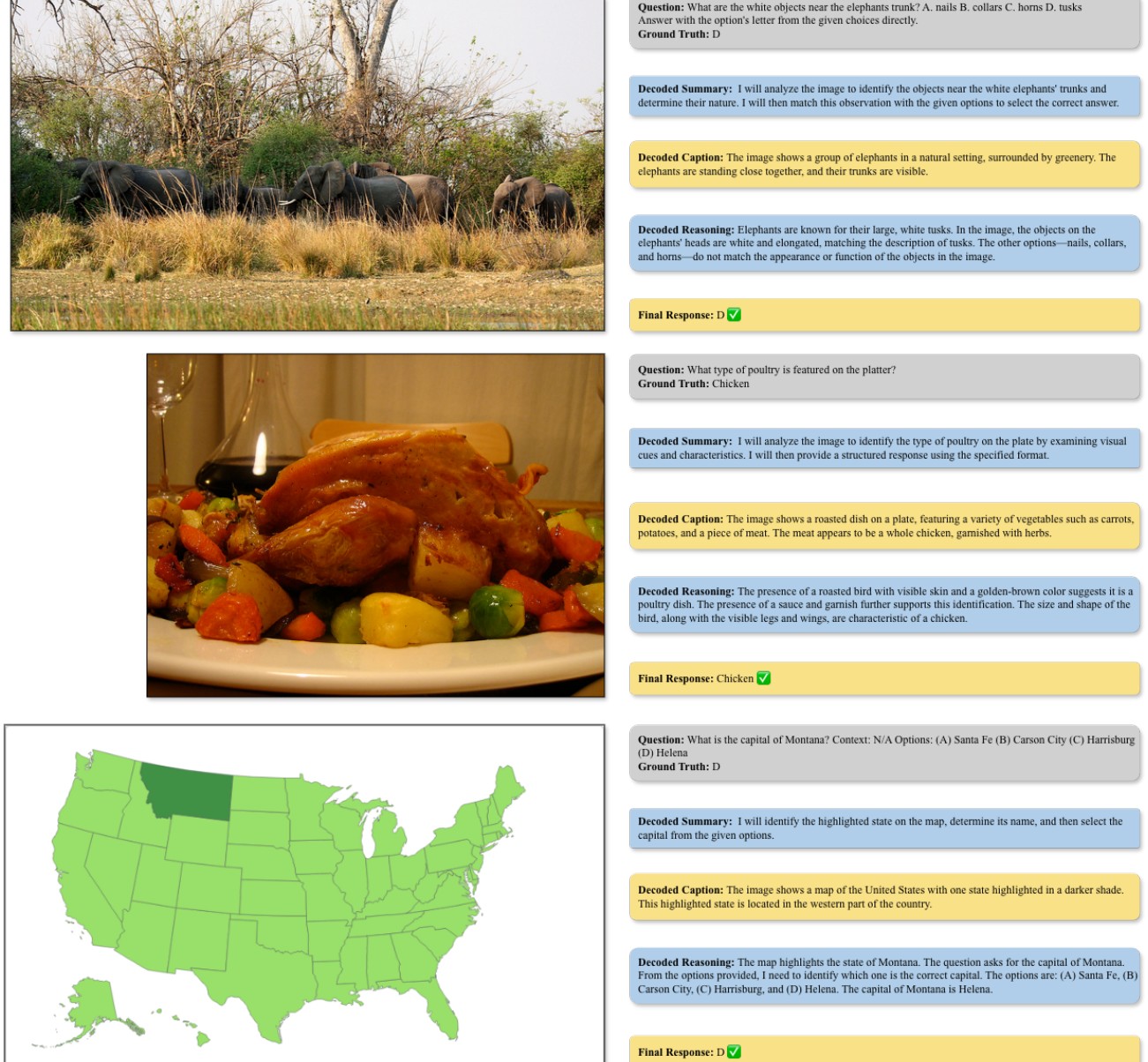

*Figure 6.* More qualitative examples demonstrating SELR's capability to decode latent thoughts into human-readable summaries, captions, and reasoning steps.

---

**Algorithm 2** GPT-4o Prompt for CoT Decoding Evaluation (from Heima (Shen et al., 2025a))

---

**Input:** Image **I**, question **Q**, decoded CoT **CôT**, ground truth **CoT**, type of CoT stage **T** $\in$ [**caption**, **summary**, **reasoning**].

**Output:** An integer representing the rank of similarity between CôT and CoT in [1, 5].

**User:** When responding to questions about an image, a deep analysis is crucial for providing accurate answers. The analysis of an image-question pair could be one of the following components:

**Summary** – A brief restatement or paraphrasing of the question.

**Caption** – A description or summary of the content of the image.

**Reasoning** – A logical explanation of how the answer is derived from the image and the question.

You will be provided with one of them along with the ground truth. Your task is to evaluate whether the analysis closely aligns with the ground truth according to the given image and question pair.

**User:** In this conversation, you will be given a generated **T** and its ground truth.
The **T** is: **CôT**.
The ground truth is: **CoT**.

**User:** Following is the given image: **I**
The corresponding question is: **Q**.

**User:** Please rank the similarity with an integer between 1 and 5, where the larger number means the generated **T** is closer to the ground truth. Please rate the similarity on a scale from 1 to 5, where:

**1: Completely unrelated.** The generated **T** and ground truth discuss entirely different themes, and there is no overlap in content, or subject matter. Example: Ground Truth: ...; Generated **T**: ...

**2: Minimally related.** The generated **T** and ground truth are tangentially connected. Only a minimum fraction of the theme or content in the ground truth is mentioned in the generated **T**. Example: Ground Truth: ...; Generated **T**: ...

**3: Somewhat related but with notable discrepancies.** The generated **T** and ground truth share key elements in theme or content but exhibit clear differences in focus, description, or details. While the overall themes or settings may overlap (*e.g.*, animals, fences, grassy area), the generated **T** introduces significant factual errors or omits important details. Example: Ground Truth: ...; Generated **T**: ...

**4: Closely related with small differences.** The generated **T** and ground truth align on the main theme and share most of the key details. However, there are minor differences in phrasing, specific details, or focus. Example: Ground Truth: ...; Generated **T**: ...

**5: Nearly identical.** The generated **T** and ground truth are highly similar, sharing nearly all content, details, and key descriptions, with only minor or negligible phrasing differences. Example: Ground Truth: ...; Generated **T**: ...

The output should be in a JSON format:
{"**T**" : (Rank), "reason" : ...}
(Rank) is the integer of the similarity rank.
"reason" stores the reason for ranking a given **T** and ground truth.

---

