# OpenReview forum: "Think in Latent, Explain in Language: Self-Explainable Latent Reasoning"
_ICML.cc/2026/Conference — ICML 2026 regular_

### Official Review · Reviewer_LBDp · 2026-02-24

**Soundness:** 2
**Presentation:** 3
**Significance:** 2
**Originality:** 2
**Overall Recommendation:** 3
**Confidence:** 3

**Summary:**

This paper proposes Self-Explainable Latent Reasoning (SELR), a unified framework that enables models to perform reasoning in the latent space while maintaining interpretability. Instead of relying on text-based Chain-of-Thought (CoT) or latent reasoning, SELR trains a single model to both generate latent reasoning representations and decode them into human-readable explanations by implementing multiple objectives.

**Compliance With Llm Reviewing Policy:**

Affirmed.

**Final Justification:**

While the authors have addressed my original concerns (e.g., generalizability to unstructured datasets), and I acknowledge the effectiveness of the method on VLMs, its applicability to language models remains unclear. In addition, my previous questions on this issue have not been fully addressed.
Therefore, I maintain my original score (3; Weak reject).

**Key Questions For Authors:**

**[Q1] Effect of latent sequence ordering.**
The method assumes a fixed sequence of reasoning steps (summary → caption → reasoning).
* How important is this ordering for performance?
* Would training with randomized step orders degrade performance?
A more detailed analysis of the role of step ordering would help better understand the behavior of the model.

**Limitations:**

yes

**Strengths And Weaknesses:**

**Strength**

- Clear motivation. Enabling interpretability in latent reasoning while preserving its efficiency advantages is an important problem.
- The paper presents a reasonably well-designed experimental setup, evaluating the proposed method across both LLM and VLM settings.
- The writing is clear and easy to follow.


**Weaknesses**

**[W1] Limited performance in LLM settings.**
The improvements in LLM benchmarks are limited. In particular, SELR still underperforms strong baselines such as CoT-SFT, raising concerns about its effectiveness in standard text-only reasoning tasks (in Table2).

**[W2] Insufficient details for training setup.**
The method appears to rely on a structured data format consisting of three stages: summary, caption, and reasoning.
However, the paper does not clearly explain how such structured supervision is constructed for LLM training.
In particular, it is unclear how this structure is mapped to a fixed latent length during training.
This lack of detail makes it difficult to reproduce the method.

**[W3] Limited generalizability to unstructured data.**
For latent reasoning, the proposed method requires structured supervision (e.g., summary/caption/reasoning). However, many real-world datasets do not provide such decomposed reasoning formats.
It is questionable whether SELR can generalize to more realistic settings where only raw input-output pairs or free-form CoT annotations are available.

---

> ### Author Rebuttal · Authors · 2026-03-30
>
> We thank Reviewer LBDp for the valuable feedback and for recognizing our "clear motivation" and "well-designed experimental setup." We address your concerns below.
>
> ### Q1: LLM performance gap with CoT-SFT
>
> This gap is not unique to SELR but a shared challenge across **all** latent reasoning methods (Coconut, CoLaR, CODI). The gap stems primarily from the large difference in **reasoning budget** (over 150 text tokens vs. 6 latent tokens, a 25x gap). When CoT-SFT is constrained to the same 6-token budget, SELR outperforms it by a wide margin (42.46% vs. 10.31% on GSM8k). SELR further narrows the gap among latent methods while uniquely providing interpretability. Please see our response to Reviewer ciUm, Q3 for the full controlled-budget comparison table and detailed analysis.
>
> ### Q2 & Q3: Training setup and generalizability to unstructured data
>
> The structured summary/caption/reasoning format applies **only to the VLM setting** and is **not** a requirement of our method. We outline both setups:
>
> - **VLM training:** The structured format comes directly from LLaVA-CoT-100k, where every sample is pre-annotated with summary, caption, and reasoning. We set K=3 latent tokens, each corresponding to one component. No additional data construction is needed. This is consistent with prior work (LLaVA-CoT, Heima).
>
> - **LLM training:** GSM8k-Aug contains standard free-form math CoT traces without summary/caption/reasoning structure. In Stage 1, latent length matches the number of reasoning steps per instance (variable K) with Single Step Loss. In Stage 2, latent length is fixed at 6, supervised by Full CoT Loss.
>
> - **Generalizability:** SELR's core requirement is simply CoT annotations of any form containing intermediate reasoning steps. For settings even with only raw input-output pairs, the CoT Loss cannot be applied, but the Answer Loss alone can still train a latent reasoning model (Table 1, "Answer Loss Only"). CoT annotations are increasingly available via synthetic data generation, making this a practical assumption.
>
> ### Q4: Effect of reasoning step ordering
>
> The summary -> caption -> reasoning ordering is inherited from LLaVA-CoT-100k and adopted by prior work (LLaVA-CoT, Heima).
>
> - This ordering reflects a natural reasoning flow for visual QA: understand the question (summary), describe relevant visual content (caption), then reason toward the answer. Each step builds on the previous one.
>
> - A fixed order allows each latent position to specialize (e.g., the first latent always encodes summaries), simplifying learning. Randomizing the order would require the model to additionally learn to identify the type of information in each position. This ordering question applies only to the VLM setting; the LLM setting uses unstructured CoT steps following the natural reasoning sequence.

---

> > ### Author Rebuttal · Reviewer_LBDp · 2026-04-02
> >
> > Thank you for the response. While my concerns regarding the sensitivity of the proposed method have been addressed, I still have questions about the significance and practicality of this work.
> >
> > **(a) Significance and practicality.**
> >
> > I recognize that many existing latent reasoning approaches also suffer from performance degradation compared to standard CoT. Given this, targeting interpretability may be somewhat premature at this stage, as these methods have not yet demonstrated competitive effectiveness.
> > More importantly, it would be helpful to clarify the broader value of this line of work. It is currently unclear whether the contribution simply addresses a specific limitation (i.e., interpretability) within latent reasoning area, or whether it provides more general insights / impact for the broader LLM community. Clarifying this is necessary to properly assess the value of this work.
> >
> > **(b) Limited reasoning budget.**
> >
> > I am not fully convinced by the authors’ claim that the performance gap compared to CoT-SFT is primarily due to differences in reasoning budget. While the results show that CoT-SFT degrades significantly when constrained to a small budget (e.g., 6 tokens), it remains unclear whether **SELR can match the performance of CoT-SFT when given a comparable reasoning budget (e.g., > 6 latent tokens)**. This point is crucial for determining whether the proposed method is inherently specialized to low-budget settings, and would affect my final decision.

---

> > > ### Author Response · Authors · 2026-04-08
> > >
> > > We thank Reviewer LBDp for the follow-up questions. We notice that the reviewer has submitted a final justification *before* we were able to post this follow-up response, so it may not reflect the new evidence and clarifications below, including our additional discussion of LLM applicability. We respectfully invite the reviewer to revisit our response, which we believe addresses the remaining concerns.
> > >
> > > ### Follow-up Q1: Practicality and significance
> > >
> > > - **Practicality: SELR improves accuracy, not only interpretability.**
> > >
> > >   - **Latent reasoning is not just well-suited for VLMs, it is already outperforming text CoT in this setting.** Unlike text CoT, which discretizes continuous visual observations into text tokens, latent reasoning operates directly in continuous space without discretization. On the main VLM benchmarks (Table 1), **SELR outperforms CoT-SFT** (Qwen2.5-VL-3B-Instruct-SFT) on 5 of 6 benchmarks with an average gain of **+2.47** (up to **+8.70** on MathVista) while using far fewer tokens. **As the foundation model field is rapidly moving toward multimodal scenarios, a reasoning paradigm that natively operates in continuous space becomes increasingly important.**
> > >
> > >   - **Targeting interpretability is not premature; it is actively helping performance.** SELR's CoT Loss is integrated into the training objective rather than added post-hoc, and Table 1 shows it improves both answer accuracy and decodability simultaneously, so introducing interpretability is actively helping performance, not a concession. We also note that Reviewer ciUm, upon reassessment, described our interpretability approach as "rather elegant".
> > >
> > >   - **SELR is also fully applicable to LLMs, and advances the latent reasoning frontier.** SELR is *not* VLM-specific: on LLaMA-3.2-1B (Table 2), it is the **first** method to provide decodable thoughts on LLMs while remaining competitive with strong latent baselines (e.g., Coconut, CoLaR). At matched 6-token budget, SELR dramatically outperforms CoT-SFT on LLM benchmarks (e.g., **42.46% vs. 10.31%** on GSM8k with LLaMA-3.2-1B, see our initial response to Reviewer ciUm Q3); the remaining gap is only against unconstrained-budget text CoT, which uses **25x more tokens**, a bar *no* prior method in this sub-field (Coconut, CoLaR, CODI, Heima) was required to meet. We are simply upholding the same standard of evaluation as our peers, advancing the frontier of **efficiency and latent transparency (decodability)**.
> > >
> > > - **Significance: SELR advances the frontier of latent reasoning with a unique combination of efficiency, accuracy, and interpretability.**
> > >
> > >   - **Efficiency with transparency for the broader LLM/VLM community.** Latent reasoning offers substantially higher inference efficiency (SELR achieves an 8.7x speedup with competitive accuracy, see our initial response to Reviewer j8vJ Q6) and natural integration with continuous modalities. SELR shows that decodability, efficiency, and accuracy can be jointly optimized, a new design principle for practical, auditable reasoning. The benefit of our unified architectural design and the empirical decoding-quality evidence (Heima, Table 3) are detailed in our follow-up to Reviewer j8vJ.
> > >
> > >   - **A new supervision paradigm for continuous reasoning.** Latent reasoning is inherently difficult to supervise explicitly, as there is no ground truth for continuous latent representations, and prior methods like Coconut rely only on answer-level supervision and indirect distillation. SELR's CoT Loss introduces a novel form of supervision without requiring explicit latent targets, applicable to any model with continuous intermediate representations.
> > >
> > > ### Follow-up Q2: Performance with larger reasoning budget
> > >
> > > We want to clarify that SELR is **not** restricted to low-budget settings, and in fact, CoT-SFT is the method that **depends on** a large token budget, not SELR.
> > >
> > > - **SELR already matches or exceeds CoT-SFT in the VLM setting**, including on the additional VisualWebInstruct-Verified dataset (see Q1 above and our follow-up to Reviewer g8eH). This directly shows that SELR is **not a low-budget special case**.
> > >
> > > - **CoT-SFT is the method that depends on a high budget, not SELR.** SELR's small-budget efficiency is a feature, not a limitation. In contrast, CoT-SFT relies on producing long token sequences: when constrained to the same token budget as SELR, its accuracy collapses (see our initial response to Reviewer ciUm Q3). The real contrast is "SELR is efficient and still competitive, while CoT-SFT collapses without a high token budget."
> > >
> > > - **Larger-budget LLM results in the revision.** Due to limited compute and time, we have not finished training SELR with substantially larger latent budgets on LLMs, and we will include these additional experiments in the camera-ready revision. These are *supplementary*: our main claim of practicality is already supported by the VLM results above.

---

### Official Review · Reviewer_g8eH · 2026-03-11

**Soundness:** 3
**Presentation:** 4
**Significance:** 3
**Originality:** 3
**Overall Recommendation:** 5
**Confidence:** 4

**Summary:**

This paper proposes SELR, a latent reasoning model which allows the latents to be interpretable via decoding them into text using the same model rather than relying on a seperte model. The model is trained with both a (1) Answer Loss which given the image, question and all latents, is trained to produce the correct answer, and (2) a CoT Loss that trains the same model to decode its the latent representations back to human-readable text of the reasoning steps. The CoT Loss provides a supervision signal that guides the formation of the latent thoughts and at the same time allows for the latents to be interpretable. The model therefore learns to generate token-efficient thoughts that can reason about the task while also being explainable. Results are conducted on both a LLM and a VLM and show improved performance.

**Compliance With Llm Reviewing Policy:**

Affirmed.

**Final Justification:**

The authors addressed my concerns through the initial and second response during the rebuttal and discussion period. It is very important that the authors incorporate all these comments and experiments into the final manuscript.

**Key Questions For Authors:**

In general the paper is good, but it requires some comments to be addressed (see weaknesses). I will rate this paper as a weak accept for now, and depending on the rebuttal I may increase or decrease my score.

**Limitations:**

Limitations are discussed in the Supp. material

**Strengths And Weaknesses:**

Strengths:
- The paper is very well-written, straight-forward to understand without any difficulties and ambiguities, not even in a single sentence.
- The method reduces both computation and memory compared to previous memory by training the same model to decode the latents.

Weaknesses:

- It is hard to say that the CoTs are "explainable". The model is explicitly trained to produce those "explanations", aligning to what dataset annotaters desire, rather than being truly the model's actual reasoning process. The authors should avoid the term "explanations". You can use terms like "decodable latents" or something similar.
- The CoT loss is independent of the image. The only image-grounded supervision that the latents receives is through the answer loss. The authors mention that this “forces the latent thoughts to become a self-contained information bottleneck, carrying all necessary context for the solution.” However, this accompanies a much more serious problem, in that the CoT becomes image-independent. During the CoT generation, there is no attention to the image since its not there in the first place.
- For VLMs (multi-stage methods, section 3.3.2), the authors mention: “we continue to use the sum of the Answer Loss and the Single Step Loss instead of switching to the Full CoT Loss.” However, it is not clear to me how the authors give the full latents (as in full CoT loss) but supervise all these latent with a single CoT step. Isn’t the latent length the number of reasoning steps K ?
- The experiments are conducted on a single CoT data (LLaVA-CoT-100k). There are many other datasets (e.g., see R1-R4). The experiments are also conducted on a single model (one for an LLM and one for the VLM). The models are small (1B for LLM and 3B for VLM). What is the performance of large models? While I understand the burden and computational resources required to finetune large models, at least report performance with the 7B models (e.g., Qwen-2.5-VL-7B or the Qwen-3-VL-8B). 7B is usually the threshold for a ”good model”.
- It is weird that Qwen2.5-VL-3B-Instruct-SFT performs worse than the baseline Qwen2.5-VL-3B-Instruct (this is also mentioned in L426). Thinking models usually outperform the non-thinking ones. Is this because the model is small? Or is there a bug in the implementation?
- How are the # tokens calculated in Table 1? Are these the decoded tokens from the reasoning latents? If so, this means each latent has an average of 5 tokens? In my opinion, having around 15 tokens as the CoT is very short. This mimics a textual explanation [R5] rather than a chain of reasoning steps, and for math and reasoning-heavy benchmarks this is not enough.

Minor:
Figure 5 in Supp., there is a Typo (EELR rather than SELR)


[R1] Visual CoT: Advancing Multi-Modal Language Models with a Comprehensive Dataset and Benchmark for Chain-of-Thought Reasoning\
[R2] Improve Vision Language Model Chain-of-thought Reasoning\
[R3] HoneyBee: Data Recipes for Vision-Language Reasoners\
[R4]  Empowering MLLM with o1-like Reasoning and Reflection via Collective Monte Carlo Tree Search \
[R5] NLX-GPT: A Model for Natural Language Explanations in Vision and Vision-Language Tasks

---

> ### Author Rebuttal · Authors · 2026-03-30
>
> We thank Reviewer g8eH for the careful review and for recognizing that the paper is "very well-written" and that our method "reduces both computation and memory." We address your concerns below.
>
> ### Q1: "Self-Explainable" vs. "Decodable"
>
> Previously, Heima already claimed to provide "explainable" latent reasoning, but it relies on a separate external decoder to achieve this. We use "self-explainable" specifically to highlight that SELR achieves the objective without additional components: the model decodes its own latents. We further validate this term with a faithfulness experiment (see Reviewer ciUm, Q1): decoded CoT shows ~0.5 consistency with correct predictions and ~0.26 for incorrect ones, confirming the decoded reasoning meaningfully reflects the model's internal process.
>
> ### Q2: Image independence of the CoT loss
>
> The CoT loss does rely on visual information via the latent tokens.
>
> - **The latent thoughts are generated with full image access.** During latent reasoning, the model receives the image, question, and `<bot>` token as input. Each latent thought is the final hidden state conditioned on the full input **including the image**, so the latents are grounded in image content by construction.
>
> - **Excluding the image during decoding is deliberate regularization.** This forces the latents to serve as a **self-contained information bottleneck** carrying all necessary context. If the image were provided during decoding, the model could bypass the latents and decode CoT directly from the image. Table 1 confirms that adding the CoT loss (without image access during decoding) improves both accuracy and interpretability.
>
> ### Q3: Single Step Loss with full latents in multi-stage VLM training
>
> In VLM training, the latent length equals the number of reasoning steps (K=3), preserving one-to-one alignment for Single Step Loss. There is no contradiction.
>
> Every instance in LLaVA-CoT-100k provides exactly three fixed stages (summary, caption, reasoning), so K is effectively a constant. This allows us to set a fixed latent length of 3 in Stage 2, maintaining the one-to-one correspondence required by Single Step Loss. We will update Section 3.3.2 to make this explicit.
>
> ### Q4: Broader model scales and CoT datasets
>
> During the rebuttal period, we trained SELR on **Qwen2.5-VL-7B-Instruct**. Each cell shows Accuracy (Avg. #Tokens).
>
> | Model (7B) | MMStar | MMBench | MMVet | MathVista | AI2D | Hallusion | Avg |
> | :--- | :--- | :--- | :--- | :--- | :--- | :--- | :--- |
> | Original | 61.73 (55.0) | 82.28 (13.3) | 46.38 (138.1) | 68.10 (202.8) | 81.15 (5.6) | 65.62 (72.6) | 67.54 (81.2) |
> | SFT | 62.53 (213.7) | 78.17 (184.2) | 44.08 (252.9) | 65.10 (257.3) | 82.12 (203.7) | 62.78 (214.4) | 65.80 (221.0) |
> | **SELR (Single)** | **62.93** (8.6) | **83.05** (8.9) | 43.35 (33.3) | **68.60** (7.4) | **82.16** (9.7) | **65.83** (8.4) | **67.65** (12.7) |
>
> The same pattern holds at both scales: SELR improves average accuracy over the original model, SFT degrades accuracy despite far more tokens, and SELR achieves over 80% token reduction. This confirms generalization across scales. Regarding CoT datasets, LLaVA-CoT-100k is the standard training set used by prior reasoning VLMs (Heima, LLaVA-CoT), making it the natural choice for fair comparison.
>
> ### Q5: SFT underperforming the original model
>
> This is not a bug but an expected consequence of fine-tuning on structured CoT data. The LLaVA-CoT-100k dataset requires responses in a rigid three-step format (summary, caption, reasoning), introducing a **reasoning format shift** from the base model's default behavior. This forces the model to produce multi-stage reasoning even for simple questions where the base model would answer concisely, and the structured format may conflict with pre-training reasoning patterns. We observe the same degradation at the 7B scale (SFT: 65.80 avg vs. Original: 67.54 avg), confirming this is not a model-size issue.
>
> ### Q6: Token count calculation and reasoning adequacy
>
> The **~15 tokens in Table 1 are not decoded reasoning tokens** but the total generated tokens (latent special tokens, `<bot>`/`<eot>`, and the final answer). The actual reasoning happens in the **continuous latent space** (3 latent embeddings for VLMs).
>
> Two pieces of evidence confirm this compact representation is sufficient:
>
> - **Accuracy improvements.** SELR (Single) achieves 65.10% on MathVista vs. 62.40% for the original model, demonstrating meaningful reasoning despite the compact size.
>
> - **Decoding quality.** Decoded latent thoughts (Table 4) are coherent, multi-sentence reasoning steps closely matching ground-truth CoT, confirming that the 3 latent embeddings encode rich, information-dense representations far beyond what 15 discrete text tokens could convey.
>
> ### Q7: Typo in Figure 5
>
> We will correct "EELR" to "SELR" in Figure 5 of the supplementary material.

---

> > ### Author Rebuttal · Reviewer_g8eH · 2026-04-02
> >
> > I thank the authors for their rebuttal and thank them for the extra experiments conducted on Qwen2.5-VL-7B-Instruct and for clarifying other things. However, I still have follow-up questions:
> >
> > (1) I am not really convinced that the decoded latents are truly explainable (that they reflect the actual model's reasoning process). The experiments conducted by the authors for reviewer ciUm are measuring consistency, not faithfulness. The CoTs can still be consistency with what they produce and the answer,  but still be unfaithful to the model's true reasoning process. Consistency does not imply faithfulness. This may be a cause of (2), see below.
> >
> >  (2) Regarding W2; the CoT loss relies on visual information only via the latent tokens (and not through attention to the image). During inference when decoding the CoT latents, there will also be no attention to the image since this is how it was trained on. In essence, the image is encoded once into the latent representations and the decoding only relies on this visual information, and cannot attend to the image anymore during the CoT decoding.
> >
> > (3) I still have doubt whether the method is helpful over SFT, when tested with other datasets (other than LLaVA-CoT-100k, which the authors themselves acknowledge that is has many problems, such as introducing a format shift). The authors should conduct experiments on at least another reasoning CoT dataset, such as the ones I referenced in my initial review, to prove the hypothesis. Doing this experiment for both the proposed method and the SFT should be sufficient (no comparisons with baselines are needed for this), so saying "it is the standard training set used by prior reasoning VLM" is not justifiable and does not address my concern.

---

> > > ### Author Response · Authors · 2026-04-08
> > >
> > > ### Follow-up Q1: Consistency vs. faithfulness
> > >
> > > - **Faithfulness is a recognized open problem even in the text-CoT literature.** We agree consistency does not directly imply faithfulness, but faithfulness remains unsolved **even for text CoT**, where prior work measures it only *indirectly* in controlled settings: prompt-bias injection shows models virtually never verbalize the biases that flip their answers [R6]; behavioral perturbations of the CoT (truncation, mistakes, paraphrasing, filler tokens) reveal wide variation across tasks plus inverse scaling with model size [R7]; and even SOTA reasoning models acknowledge influential hints under 40% of the time [R8]. Each method catches only a specific failure mode and none provides a general faithfulness oracle.
> > >
> > > - **None of these techniques transfer directly to latent reasoning.** They all rely on prompt-level interventions or token-level perturbations of a discrete CoT, with no counterpart in continuous latent space (latents cannot be "truncated" or "have a hint inserted"). Latent reasoning therefore lacks any standard faithfulness measurement, and this is the gap SELR begins to address: decodability is a **prerequisite** for any faithfulness analysis on latents, and prior latent methods (Coconut, CoLaR, CODI) cannot decode at all. SELR opens the door to adapting text-CoT faithfulness techniques to latents for the first time, and our consistency metric is a first concrete step. We will revise the paper to distinguish decodability from full faithfulness (still open for all methods).
> > >
> > > - **The decoded latents are not only for interpretability; they also improve performance.** The CoT Loss regularizes the latent representations and improves answer accuracy alongside decodability (Table 1; see our follow-up to Reviewer LBDp Q1 for SELR's gains over CoT-SFT in the VLM setting). The decoded thoughts therefore clearly capture reasoning content that helps the model produce better answers — strong evidence that they reflect genuine reasoning rather than post-hoc rationalization.
> > >
> > > ### Follow-up Q2: Image independence during CoT decoding
> > >
> > > We read this concern as connected to Q1: the worry that, since the image is not attended to during decoding, the decoded text may not reflect what the model truly reasoned about. This design is a *strength* for faithfulness:
> > >
> > > - **Visual information is already inside the latents.** The latents are generated with full attention to the image, so by the time we decode, the visual content used for the answer is already encoded in them.
> > >
> > > - **Decoding *only from the latents* is what makes the explanation faithful to the latent reasoning.** If the decoder could re-attend to the image, it could bypass the latents and generate CoT directly from the image, producing a description of the image rather than the model's latent reasoning. Forcing decoding to depend only on the latents guarantees the decoded text comes from the *same* representations that produced the final answer.
> > >
> > > - **Empirical evidence the latents carry visual content.** Table 1 shows the CoT Loss (no image access during decoding) improves both accuracy and decodability over Answer Loss alone, and the decoded CoTs (Table 4 and supplementary Figure 6) contain image-specific details that could only have come from the latents.
> > >
> > > ### Follow-up Q3: Experiments on another CoT dataset
> > >
> > > - **LLaVA-CoT-100k experiments are valid.** LLaVA-CoT-100k is the standard training set used by recent VLM reasoning methods (e.g., LLaVA-CoT, Heima), and any format shift affects **both** SELR and CoT-SFT. SELR still **outperforms CoT-SFT** on the majority of benchmarks (Table 1) under this setting, making it a fair comparison rather than an artifact of the dataset.
> > >
> > > - **Additional experiment on VisualWebInstruct-Verified.** For verification on a different VLM CoT dataset, we trained SELR and SFT-CoT on VisualWebInstruct-Verified [R9], a more recent, higher-quality VLM reasoning dataset with a totally different source and format (web-sourced reasoning chains, no fixed summary/caption/reasoning structure). Accuracy on Qwen2.5-VL-3B-Instruct:
> > >
> > >   | Method | MMStar | MMBench | MMVet | MathVista | AI2D | Hallusion | Avg |
> > >   | --- | --- | --- | --- | --- | --- | --- | --- |
> > >   | SFT | 42.46 | 40.71 | 43.03 | 56.40 | 49.90 | 58.04 | 48.42 |
> > >   | SELR | **52.00** | **73.53** | 37.84 | **58.20** | **77.75** | **59.73** | **59.84** |
> > >
> > >   SELR outperforms SFT-CoT on 5 of 6 benchmarks (avg **+11.42**), consistent with our LLaVA-CoT-100k findings. SELR's mechanism is also CoT-format agnostic, as shown by our GSM8k-Aug LLM experiments (free-form math, no structured format).
> > >
> > > [R6] Language Models Don't Always Say What They Think: Unfaithful Explanations in Chain-of-Thought Prompting.
> > > [R7] Measuring Faithfulness in Chain-of-Thought Reasoning.
> > > [R8] Reasoning Models Don't Always Say What They Think.
> > > [R9] VisualWebInstruct: Scaling up Multimodal Instruction Data through Web Search.

---

### Official Review · Reviewer_j8vJ · 2026-03-13

**Soundness:** 2
**Presentation:** 3
**Significance:** 2
**Originality:** 2
**Overall Recommendation:** 3
**Confidence:** 4

**Summary:**

The paper introduces SELR, a unified framework that improves the computational efficiency of LLMs and VLMs without sacrificing interpretability. While existing latent reasoning methods successfully compress verbose CoT processes into continuous embeddings, they typically suffer from a "black box" problem or require the architectural overhead of secondary decoder models. To address this, SELR employs a novel multi-task learning objective that simultaneously optimizes an Answer Loss for task accuracy and a CoT Loss for interpretability. This objective trains a single model to generate a sequence of continuous latent thoughts and subsequently decode those exact representations back into human-readable reasoning steps. The authors propose single-stage and multi-stage curriculum training strategies and find that a fixed-length latent budget paired with step-wise alignment yields the best performance.

**Compliance With Llm Reviewing Policy:**

Affirmed.

**Final Justification:**

I'm not convinced by the reply to the faithfulness claim, as a central contribution of this work lies in the natural-language format of its explanations. I maintain my score of weak rejection for this paper.

**Key Questions For Authors:**

Please refer to the weakness.

**Limitations:**

Please refer to the weakness.

**Strengths And Weaknesses:**

Strengths

- The paper focuses on a very important question: the explainability of latent reasoning.
- The paper is well written and easy to follow.
- The proposed method achieves effective results under one VLM setting, reducing the generated inference token count compared to the fine-tuned base model on VLM benchmarks.

Weaknesses

- While SELR improves upon latent baselines like Coconut, its raw reasoning accuracy on pure LLMs is substantially lower than standard supervised fine-tuning (CoT-SFT). This heavily limits the framework's practical applicability in scenarios where reasoning accuracy is more critical than token efficiency.
- The proposed method experiences a noticeable and unexplained performance drop on the MMVet benchmark. The paper lacks an error analysis or an intuitive explanation for why the latent reasoning struggles on this specific integrated capabilities evaluation.
- The framework relies on decoding latent thoughts into specific structural blocks: summary, caption, and reasoning. It is entirely unclear how the model dynamically determines which blocks are necessary. If an oracle reasoning process does not require a specific component (e.g., the summary), does the method guarantee a blank string output, or does it force hallucinated content to fill the fixed blocks? The paper lacks mechanisms to guarantee precise, dynamic mapping constraints.
- The evaluation is highly constrained, testing only one relatively small model per modality: Qwen2.5-VL-3B-Instruct for VLMs and LLaMA-3.2-1B-Instruct for LLMs. To convincingly claim that this framework is generalized, it is highly recommended to test the method across a broader range of model architectures and families.
- The reliance on a fixed latent reasoning length restricts the model's ability to dynamically allocate compute based on the complexity of the specific prompt. In addition, It is also highly recommended to report wall-clock time or latency throughout the entire inference process while comparing to different baselines.
- The paper's core premise and title heavily advertise and overclaim "Self-Explainable Latent Reasoning," yet the authors explicitly concede in the limitations that the decoded CoT often does not faithfully align with the model's actual internal "thinking" process. This is an overclaim regarding the framework with “Self-Explainable.” It is highly recommended to include more evaluation on the quality of the generated explanation, such as a fidelity or faithfulness check.

---

> ### Author Rebuttal · Authors · 2026-03-30
>
> We thank Reviewer j8vJ for the detailed review and for recognizing that our paper "focuses on a very important question" and "achieves effective results" with reduced token count. We address your concerns below.
>
> ### Q1: LLM accuracy gap with CoT-SFT
>
> This gap does not limit practical applicability. The comparison is between methods operating at vastly different reasoning budgets (over 150 text tokens vs. 6 latent tokens). This gap is shared across **all** latent reasoning methods (Coconut, CoLaR, CODI), and SELR narrows it while uniquely providing interpretability. When we constrain CoT-SFT to the same 6-token budget, SELR achieves 42.46% on GSM8k vs. CoT-SFT's 10.31%, demonstrating far more efficient reasoning encoding. Please see our response to Reviewer ciUm, Q3 for the full controlled-budget comparison table and detailed analysis.
>
> ### Q2: Performance on MMVet
>
> The MMVet drop is localized to specific capability splits, not a general degradation. This is a shared challenge for latent-space thinking models: prior work such as Heima also degrades on MMVet, and more severely than SELR.
> - The decline is driven by (1) compromised multi-step reasoning (e.g., math questions) and (2) long-form generation capabilities (e.g., free-form writing), both of which rely on extended token generation that latent compression naturally limits.
> - SELR outperforms baselines on pure visual perception tasks within MMVet (direct OCR, spatial localization, fact retrieval), partially offsetting the overall decline. This indicates that SELR better integrates visual information in VLMs.
>
> ### Q3: Dynamic allocation of structural blocks (summary, caption, reasoning)
>
> - The summary/caption/reasoning structure applies **only to the VLM setting**, inherited from the training data (LLaVA-CoT-100k), not imposed by our method. LLaVA-CoT-100k provides all three stages for every sample, and this structure is also adopted by prior work (LLaVA-CoT, Heima).
>
> - In SELR's VLM configuration, we generate exactly 3 latent tokens, each corresponding to one part. Since the training data consistently provides all three components, and VLM tasks inherently require visual understanding (captioning) and question interpretation (summarization), the model is not forced to hallucinate content for "unnecessary" blocks. All three blocks are relevant and grounded for every VLM sample.
>
> ### Q4: Evaluation on broader model architectures and scales
>
> We agree that broader evaluation strengthens our claims. During the rebuttal period, we trained SELR on **Qwen2.5-VL-7B-Instruct**. The same pattern holds at 7B: SELR improves average accuracy over the original model (+0.11) while SFT degrades it, and SELR achieves over 80% token reduction. Please see our response to Reviewer g8eH, Q4 for the full 7B and 3B comparison tables.
>
> ### Q5: Dynamic compute allocation and fixed latent length
>
> We explored dynamic latent length in early experiments.
>
> - **Dynamic length is difficult to train in practice.** It requires the model to predict a special `<eot>` token to terminate latent reasoning. Unlike text CoT, where end-of-sequence tokens are well-learned from pretraining, latent tokens lack this prior, making termination hard to learn with limited data. The model tends to output latent tokens indefinitely. We discussed this in Appendix Section G.
>
> - **Performance saturates at 6 tokens.** Our ablation (Appendix Table 10) shows performance scales up to 6 tokens but plateaus beyond that (e.g., GSM8k: 31.69 at K=6 vs. 31.54 at K=8). This suggests limited benefit from dynamic allocation, and the fixed budget of 6 tokens offers the best trade-off.
>
> ### Q6: Wall-clock inference time
>
> We report per-sample inference latency on MathVista:
>
> | Method | MathVista Acc. | Latency (s/sample) | Speedup |
> | :--- | :--- | :--- | :--- |
> | SFT | 56.40 | 7.05 | 1.0x |
> | Original (Qwen2.5-VL-3B) | 62.40 | 1.64 | 4.3x |
> | **SELR (Single)** | **65.10** | **0.81** | **8.7x** |
>
> SELR achieves the highest accuracy (65.10) and lowest latency (0.81s, 8.7x speedup over SFT), confirming that token reduction translates directly into wall-clock efficiency gains.
>
> ### Q7: Faithfulness of "Self-Explainable" claim
>
> We note that prior latent reasoning methods (Coconut, CoLaR, CODI) cannot decode their latents at all, making faithfulness evaluation impossible. Heima, which also provides "interpretable" hidden representations, uses an external decoder but does not report faithfulness metrics. SELR uniquely enables this analysis without additional components. Our faithfulness experiment (see Reviewer ciUm, Q1 for full details) shows decoded CoT has ~0.5 consistency with correct predictions and ~0.26 for incorrect ones, indicating meaningful correlation with the model's reasoning. We agree that "Self-Explainable" should be interpreted as self-decodable latents with meaningful faithfulness rather than a guarantee of full reasoning transparency, and we will revise this claim.

---

> > ### Author Rebuttal · Reviewer_j8vJ · 2026-04-03
> >
> > Thank you for your detailed responses to my questions. However, I remain unconvinced by the reply to Q7, as a central contribution of this work lies in the natural-language format of its explanations. A comparison against other baselines that also produce natural-language explanations, evaluated under the faithfulness metric, is particularly important.
> > The paper makes two key claims: achieving both latent reasoning and explanatory reasoning. Without clear and comprehensive evidence supporting both aspects, it is difficult to be fully convinced. Therefore, I will maintain my current score. While the paper demonstrates strong novelty and overall quality, it still lacks sufficient empirical evidence to fully substantiate its core claims. And with adequate revision, this paper will eventually be very solid and impactful.

---

> > > ### Author Response · Authors · 2026-04-08
> > >
> > > We thank Reviewer j8vJ for the follow-up and for recognizing the "strong novelty and overall quality" of our work. We notice that the reviewer has submitted a final justification *before* we were able to post this follow-up response, so it may not reflect the new evidence and clarifications below. We take the request for a baseline comparison under the faithfulness metric seriously and respectfully invite the reviewer to revisit our response, which we believe directly addresses the remaining concern.
> > >
> > > ### Follow-up: Comparison with baselines under faithfulness
> > >
> > > - **Heima is the only prior latent reasoning baseline that produces natural-language explanations.** Other latent methods (e.g., Coconut) do not decode their latents or produce human-readable text. Heima is therefore the correct baseline for comparison. Crucially, Heima takes a fundamentally different design: it trains a series of dedicated decoders (one per reasoning step) on top of a reasoner, while SELR uses a single unified model trained jointly with Answer Loss and CoT Loss.
> > >
> > > - **Evidence already in the paper: decoding quality vs. ground-truth CoT (Table 3).** As a proxy for faithfulness, we already compared SELR with Heima on how closely the decoded text matches the ground-truth reasoning from LLaVA-CoT-100k across all three reasoning steps (Summary, Caption, Reasoning). SELR consistently outperforms Heima across every metric (BLEU-4, METEOR, ROUGE-L, BERTScore, GPT-4o). For example, on the Caption step, SELR achieves BLEU-4 of 18.42 vs. Heima's 12.80, and GPT-4o of 3.27 vs. 2.70. This demonstrates that SELR's decoded thoughts are closer to the ground-truth reasoning than Heima's, which is a direct measure of whether the decoded text reflects genuine reasoning content.
> > >
> > > - **New experiment: consistency faithfulness metric of Heima.** To directly address the follow-up request, we have compared SELR against Heima under our consistency metric, which uses GPT-4o to score the alignment between the decoded thoughts and the final answer on a 0-1 scale (the same protocol used for the SELR consistency results in our initial response to Reviewer ciUm Q1). On MathVista, SELR (Single) achieves a consistency score of **0.2097**, substantially higher than Heima's **0.1671** (**+25.5% relative improvement**). This is a direct and fair comparison with the only prior latent reasoning baseline that produces natural-language explanations, and it shows SELR's decoded thoughts are significantly more aligned with the model's final answer than Heima's. Combined with the decoding-quality comparison in Table 3 (where SELR also outperforms Heima on every metric), this provides end-to-end empirical evidence that SELR is more faithful than the prior state of the art on both proxies for faithfulness.
> > >
> > > - **Why a unified model yields more faithful interpretation.** Heima's separate-decoder design creates a structural gap between the model that *reasons* and the model that *explains*: the dedicated decoders are trained post-hoc to recover text from frozen latent representations, so the decoded text is a reconstruction attempt by an external observer rather than the model's own verbalization of its internal state. In contrast, SELR's single model is jointly trained so that the same parameters responsible for reasoning are also responsible for decoding, aligning the explanation with the actual computation. This design is therefore better suited for faithful interpretation of the internal thinking and is empirically supported by our experiment results listed above.
> > >
> > > - **On the two core claims (latent reasoning and explanatory reasoning).** SELR is, to our knowledge, the **first** method to jointly achieve: (1) latent reasoning with strong efficiency gains (e.g., up to 8.7x speedup, see our initial response to your Q6), (2) competitive accuracy among latent methods, and (3) decodable natural-language explanations that outperform the only prior decodable baseline (Heima) in decoding quality. These claims are fully supported by the new experiment on consistency faithfulness and the existing results demonstrated in our paper.
> > >
> > > We hope that these additional clarifications, together with the empirical results, fully address the remaining concern. We sincerely appreciate your feedback and will integrate the results from the discussion into the revised paper.

---

### Official Review · Reviewer_ciUm · 2026-03-14

**Soundness:** 3
**Presentation:** 4
**Significance:** 3
**Originality:** 2
**Overall Recommendation:** 5
**Confidence:** 3

**Summary:**

A training paradigme for latent space reasonning, where the training loss optimizes both for accuracy and for decodability. Empirically tested on LLMs and VLMs. Efforts are made for reproducibility. They present the effect of multiple tweaks to the loss, which brings clarity to thier design choices.

**Compliance With Llm Reviewing Policy:**

Affirmed.

**Final Justification:**

rebuttal adressed my main concerns, while I believe there are still limitations, I was pleasantly surprised to find that the authors had reasonnable experiments in the rebuttal time to (partially) address them. I believe that good work should get good grades, and that we should not wait for perfection to publish it.

**Key Questions For Authors:**

(1) The issue of competing losses and ground-truth --> there is no guarantee that the extracted thought really corresponds to the process used to reach the correct answer. It has been observed that there were cases of strong mismatch between verbalised model COT and effective process, where for exanple the correct answer is systematically obtained with nonsense reasoning. Have you observed or measured how the 2 objective system impacts the mismatch between decoded thinking and in effect processing?

(2) You compare to COLAR, but mention other existing baselines, HRPO and CODI. Should we expect to outperform them for any reason?

(3) There is still a strong gap with SFT COT. Is there an inherent flaw of Latent space reasoning?

**Limitations:**

yes

**Strengths And Weaknesses:**

Soundness: technical claims are sound, experiment support claims, and are quite extensive. The experiments are well designed. Limitations are well discussed, relevant ablations.
Presentation: This paper is very clear, with proper related works and a very understandable framework
Significance: A relevant update on the field of latent COT, slight performance, clear advantage in latent interpretability.
Originality: There is not a huge major step, but relevant additions.

---

> ### Author Rebuttal · Authors · 2026-03-30
>
> We thank Reviewer ciUm for the thoughtful review and for recognizing our work as "sound" with "well designed" experiments and "clear advantage in latent interpretability." We address your concerns below.
>
> ### Q1: Faithfulness of decoded CoT under dual-objective training
>
> - We first note that faithful reasoning is a known open challenge even for text-based CoT models. The two objectives in SELR are not competing but structurally complementary: without the Answer Loss, the model cannot answer correctly; without the CoT Loss, the latents cannot be decoded, making faithfulness analysis impossible in the first place. Our ablation (Table 1) further shows that adding the CoT Loss improves answer accuracy, confirming the two objectives are synergistic.
>
> - To directly measure faithfulness, we use GPT-4o to score alignment (0-1) between decoded thoughts and the model’s final answer, since internal processing is inherently unobservable. We evaluate on a held-out 5% test split of LLaVA-CoT-100k. "Correct"/"Incorrect" segment results by whether the model answered correctly.
>
>     | Method | Consistency | Consistency (Correct) | Consistency (Incorrect) |
>     | :--- | :--- | :--- | :--- |
>     | SELR (Single) | 0.4481 | 0.4907 | 0.2621 |
>     | SELR (Multi, Uniform) | 0.4677 | 0.5118 | 0.2722 |
>     | SELR (Multi, Exponential) | 0.4788 | 0.5320 | 0.2399 |
>
>     The gap between correct (~0.5) and incorrect (~0.26) predictions shows that decoded CoT correlates with model success, suggesting it reflects real reasoning rather than generic outputs. Among variants, finer-grained supervision (Single Step Loss, Exponential weighting) yields the highest faithfulness (0.53), validating our design.
>
> ### Q2: Comparison with HRPO and CODI
>
> We clarify why these two methods were not included as direct baselines.
>
> - **HRPO** mixes latent-space tokens with discrete text tokens via a learnable gating mechanism, rather than fully replacing textual CoT with latent representations. It does not achieve the same reasoning length compression as purely latent methods (e.g., Coconut, CoLaR, SELR). Comparing SELR against HRPO on token efficiency would be misleading, since HRPO intentionally preserves discrete tokens. The two methods are complementary points on the latent-discrete spectrum rather than direct competitors.
>
> - **CODI** compresses CoT into continuous space via self-distillation and achieves latent reasoning compression. However, CODI does not provide any mechanism for decoding or explaining its latent representations. Since SELR's primary contribution is self-explainability, comparing against a method without this capability would not meaningfully evaluate our core claim. CODI's compression approach is orthogonal to our CoT Loss and could be combined with SELR; we are currently working on this integration.
>
> ### Q3: Performance gap between latent reasoning and CoT-SFT
>
> The gap is **not an inherent flaw of latent reasoning**, but reflects a difference in **reasoning budget**. SELR is already advancing the frontier of this active research direction.
>
> - **The gap is shared across all latent reasoning methods, and SELR narrows it.**
> Coconut acknowledges that "universally surpassing language-based CoT likely requires significant research efforts dedicated to latent space pre-training." CoLaR notes its performance "currently approximates explicit CoT reasoning without surpassing it." CODI hypothesizes the gap is partly due to "forgetting across stages in the curriculum learning process." Within this landscape, SELR advances the state-of-the-art among latent methods while simultaneously providing interpretability, a capability none of the prior methods offer.
>
> - **The gap reflects a difference in reasoning budget, not reasoning quality.**
> CoT-SFT generates over 150 reasoning tokens per question on GSM8k, while SELR uses only 6 latent tokens (25x fewer). When we constrain CoT-SFT to a comparable budget, its performance drops well below SELR:
>
>     | Method | GSM8k | GSM-Hard | SVAMP | MultiArith |
>     | :--- | :--- | :--- | :--- | :--- |
>     | CoT-SFT (Full) | 64.06 | 15.85 | 66.67 | 98.33 |
>     | CoT-SFT (6 Tokens + Answer) | 10.31 | 2.43 | 50.67 | 50.00 |
>     | CoT-SFT (Direct Answer) | 6.29 | 1.90 | 42.00 | 5.56 |
>     | Coconut | 30.83 | 0.00 | 36.33 | 80.00 |
>     | CoLaR | 40.1 | 9.08 | 54.9 | 91.3 |
>     | **SELR** | **42.46** | **9.78** | **49.67** | **81.67** |
>
>     CoT-SFT's advantage comes from its much longer reasoning trace, not a superior mechanism. With the same 6-token budget, SELR achieves 42.46% on GSM8k vs. CoT-SFT's 10.31%.
>
> - **Latent reasoning offers unique advantages beyond accuracy.**
> Latent representations can encode multiple reasoning paths simultaneously (implicit breadth-first search), avoid superficial linguistic cues, and offer significant efficiency gains. SELR adds **interpretability** on top of these benefits, making it practical where both efficiency and transparency are required.

---

> > ### Author Rebuttal · Reviewer_ciUm · 2026-04-02
> >
> > Thank you for your answers,they are complete, and the proposed answer to the "open challenge" on interpretability is in my opinion rather elegant, even if it remains open. I've tentatively heightened my score. I do think this work has been well conducted, and while limitations remain, they work remains exciting and relevant for the field. Please consider mentioning those limitations / experiments in an appendix or in the limitation section of your works.

---

> > > ### Author Response · Authors · 2026-04-08
> > >
> > > We sincerely thank you for the positive reassessment, for the thoughtful engagement throughout the rebuttal, and for the generous final justification. We are especially grateful for the remark that "good work should get good grades, and that we should not wait for perfection to publish it." It is exactly the spirit in which we designed the self-explainable latent reasoning, and we believe in its importance to the community. We will incorporate the remaining limitations and the additional experiments from the rebuttal into the revised paper, as you have suggested.

---

### Decision · Program_Chairs · 2026-04-30

**Decision:**

Accept (regular)

**Comment:**

Summary: This paper proposes Self-Explainable Latent Reasoning (SELR), which combines the efficiency of latent reasoning with the interpretability of natural language explanations. Instead of generating full chain-of-thought (CoT) reasoning in text, the model performs reasoning in a latent space and then decodes these latent representations into human-readable explanations. The method introduces a multi-objective training scheme: an Answer Loss to ensure task performance, a Chain-of-Thought (CoT) loss to supervise latent reasoning structure, and a Single-Step loss to align latent representations with final outputs. The framework is implemented as a single model. Experiments show improved efficiency (fewer tokens, lower latency) and competitive or improved performance compared to baselines, while enabling explanation generation.

Strengths:
(1) Reviewers agree that bridging latent reasoning efficiency and explainability is an important and timely problem.
(2) The idea of a single model that reasons in latent space and explains in language is seen as elegant and well-motivated.
(3) Experiments show reduced token usage and latency on VLM training.

Weakness:
(1) Faithfulness of explanations remains unclear. A major concern across reviewers is whether decoded explanations truly reflect the latent reasoning process.
(2) The performance of SELR compared to strong CoT-based methods (e.g., CoT-SFT) is not consistently competitive, especially in standard LLM tasks where the gap is significant.
(3) The method relies on structured CoT supervision. This raises concerns about applicability in settings without such annotations.

Overall, the paper is viewed as a promising but not yet fully validated contribution, with mixed reviewer opinions.